# Tight regulation of a nuclear HAPSTR1-HUWE1 pathway essential for mammalian life

David R Amici[1,2,3,4], Sammy Alhayek[1,2,3], Austin T Klein[1,2,3], Yi-Zhi Wang[5], Anika P Wilen[5], Weimin Song[6], Pei Zhu[1,2,7], Abhishek Thakkar[1,2,7], McKenzi A King[1,2,3], Adam WT Steffeck[1,2,7], Milad J Alasady[1,2,3,4], Clara Peek[1,2,7], Jeffrey N Savas[5], Marc L Mendillo[1,2,3]

The recently discovered HAPSTR1 protein broadly oversees cellular stress responses. This function requires HUWE1, a ubiquitin ligase that paradoxically marks HAPSTR1 for degradation, but much about this pathway remains unclear. Here, leveraging multiplexed proteomics, we find that HAPSTR1 enables nuclear localization of HUWE1 with implications for nuclear protein quality control. We show that HAPSTR1 is tightly regulated and identify ubiquitin ligase TRIP12 and deubiquitinase USP7 as upstream regulators titrating HAPSTR1 stability. Finally, we generate conditional Hapstr1 knockout mice, finding that Hapstr1-null mice are perinatal lethal, adult mice depleted of Hapstr1 have reduced fitness, and primary cells explanted from Hapstr1-null animals falter in culture coincident with HUWE1 mislocalization and broadly remodeled signaling. Notably, although HAPSTR1 potently suppresses p53, we find that Hapstr1 is essential for life even in mice lacking p53. Altogether, we identify novel components and functional insights into the conserved HAPSTR1-HUWE1 pathway and demonstrate its requirement for mammalian life.

## Introduction

Cells contain a vast array of signaling programs, which are activated in response to changing environmental conditions or failures in normal quality control processes (Galluzzi et al, 2018). These programs, collectively referred to as stress response pathways, sense a challenge to homeostasis and invoke an adaptive response. Imbalances between stress responses and the physiological stress environment are implicated in many common disease states, ranging from cancer to neurodegeneration. As such, central mechanisms by which cells titrate multiple specialized stress pathways are of great interest both as therapeutic targets and as windows into a fundamental component of cell biology.

To this end, we recently used a functional genomics approach to search for central mechanisms of stress network control (Amici et al, 2022). We identified a previously unstudied gene, HAPSTR1 (formerly: C16orf72), which functions broadly to titrate stress response signaling and cellular resilience (Amici et al, 2022). Specifically, HAPSTR1 suppresses certain growth-restrictive stress responses, in particular, that mediated by master tumor suppressor p53, while augmenting classically fitness-promoting pathways such as the oxidative stress, paracrine, and heat shock responses (Benslimane et al, 2021; Amici et al, 2022; Lü et al, 2022 Preprint; Monda et al, 2023). Accordingly, loss of HAPSTR1 in cells (Benslimane et al, 2021; Amici et al, 2022, 2023; Lü et al, 2022 Preprint; Monda et al, 2023) or worms (Amici et al, 2022) impairs fitness and adaptability.

HAPSTR1 is a ubiquitously expressed nuclear protein, with two primary isoforms (long/canonical and short/missing C-terminus) that share a conserved HBO (**H**UWE1-**b**inding and **o**ligomerization) domain (Amici et al, 2022). The HBO domain constitutes two tandem degenerate amphipathic helices and contains several residues perfectly conserved through worm, yeast, and plant HAPSTR orthologs (Amici et al, 2022). These residues mediate either HAPSTR1 dimerization (G119) or HAPSTR1 binding to the giant ubiquitin ligase HUWE1 (F90, A94, Y101) (Amici et al, 2022). As suggested by the perfect conservation of these residues, the interaction between HAPSTR1 and HUWE1 is conserved and essential for HAPSTR1 function (Amici et al, 2022). However, the nature of HAPSTR1-HUWE1 cooperation remains unclear. Intriguingly, HUWE1 robustly marks HAPSTR1 for degradation, potentially representing a feedback mechanism (Amici et al, 2022; Monda et al, 2023). The only other known component of the HAPSTR1 pathway is HAPSTR2, a tissue-specific HAPSTR paralog that binds to and stabilizes HAPSTR1 (Amici et al, 2023).

[1]Department of Biochemistry and Molecular Genetics, Northwestern University Feinberg School of Medicine, Chicago, IL, USA    [2]Simpson Querrey Center for Epigenetics, Northwestern University Feinberg School of Medicine, Chicago, IL, USA    [3]Robert H. Lurie Comprehensive Cancer Center, Northwestern University Feinberg School of Medicine, Chicago, IL, USA    [4]Medical Scientist Training Program, Northwestern University Feinberg School of Medicine, Chicago, IL, USA    [5]Department of Neurology, Northwestern University Feinberg School of Medicine, Chicago, IL, USA    [6]Comprehensive Metabolic Core, Northwestern University Feinberg School of Medicine, Chicago, IL, USA    [7]Department of Medicine, Division of Endocrinology, Metabolism, and Molecular Medicine, Northwestern University Feinberg School of Medicine, Chicago, IL, USA

Correspondence: mendillo@northwestern.edu

Here, we provide insight into many open questions about the HAPSTR1 protein. Leveraging unbiased proteomic and transcriptomic experiments, we propose that HAPSTR1 enables a non-canonical nuclear activity of HUWE1 critical for nuclear protein quality control. We identify new members of the HAPSTR1 pathway, deubiquitinase USP7 and ubiquitin ligase TRIP12, which serve as key upstream enzymes titrating HAPSTR1 and overall pathway function. Finally, we generate HAPSTR1 conditional knockout mice, demonstrating severe consequences for pathway disruption in vivo including p53-independent perinatal lethality after germline Hapstr1 loss and reduced fitness after induced Hapstr1 depletion in adults.

## Results

### HAPSTR1 is not a cofactor for canonical HUWE1-mediated ubiquitination

The remarkably similar phenotypes of HAPSTR1 and HUWE1 loss at the level of cellular fitness and transcriptome-wide signaling (Amici et al, 2022), as well as their epistatic relationship and highly conserved physical interaction (Amici et al, 2022), suggested that HAPSTR1 may be involved in HUWE1-mediated ubiquitination of a key substrate (or class of substrates). To test this hypothesis rigorously, we leveraged the relatively short half-lives of HAPSTR1 and HUWE1, which facilitate their acute depletion by siRNA within 24–30 h (Fig 1A); the model cancer cell line U2OS, which demonstrates severe growth and signaling consequences after longer term loss of either HAPSTR1 or HUWE1 (Amici et al, 2022); and quantitative, multiplexed tandem mass tag (TMT) total and diglycine proteomics (Fig 1B). Cells were harvested after 24 h of knockdown and 6 h of proteasome inhibitor (MG-132) treatment (allowing accumulation of polyubiquitinated proteins). The same cell lysates were then used for diglycine (ubiquitin) proteomics, as well as total proteomics.

Analysis of the overall proteomic landscape after acute HAPSTR1 or HUWE1 depletion revealed a shared set of proteins regulated nearly identically by loss of either factor (Fig 1C). Enriched among proteins increased after HAPSTR1 or HUWE1 loss were proteins involved in DNA replication and DNA damage responses (Fig 1C). Proteins less abundant after each knockdown are primarily related to cell cycle progression (e.g., E2F targets). Although signaling changes for HUWE1 and HAPSTR1 loss at later timepoints often reflect widespread changes (Amici et al, 2022), this more specific set of altered proteins suggests an acute defect from pathway impairment, which impacts progression through cell cycle checkpoints.

We next analyzed differential ubiquitination across 29,336 detected ubiquitination events in the same cell lysates. Despite similarly regulating the overall abundance of many proteins, we found striking discordance between HUWE1 and HAPSTR1 when assessing protein ubiquitination. Specifically, HUWE1 loss resulted in significant under-ubiquitination of many proteins at specific lysines—including proteins previously implicated as HUWE1 substrates, such as CDC6 and ribosomal proteins (Fig 1D). However, HAPSTR1 loss did not provoke loss of protein ubiquitination of

HUWE1 targets or in general (Fig 1E). These data, using the current gold-standard high-throughput approach to assess ubiquitination, implicate a host of novel proteins as putative HUWE1 substrates (Figs 1F and S1A–E), including disease-relevant proteins such as TARDBP/TDP-43. However, no strong HUWE1-dependent ubiquitination events appeared HAPSTR1-dependent, despite similar effects on absolute protein abundance in the same cell lysates (Figs 1F and S1A–E).

We note that HAPSTR1 loss did result in increased levels of several ubiquitinated peptides (Fig 1E). However, this was typically multiple lysines within the same protein, suggesting that these changes reflect changes in overall protein abundance. This was confirmed for many targets, which were also identified in total proteomics (Fig 1E). We also considered whether these proteins may relate to HUWE1. However, the vast majority did not appear to be ubiquitinated by or otherwise functionally related to HUWE1 in our datasets. Finally, we investigated whether we could identify global alterations in polyubiquitin linkages (i.e., ubiquitination sites on the ubiquitin peptide). Although we detected some subtle changes (e.g., a slight relative decrease in K33 polyubiquitination after siHUWE1), no major changes nor obvious effects of HAPSTR1 were noted (Fig S1F and G).

Altogether, these data indicate that acute HAPSTR1 loss and HUWE1 loss have immediate and shared consequences for the proteome, but that HAPSTR1 appears dispensable for HUWE1-dependent ubiquitination of specific substrate lysines.

### HAPSTR1 enables a non-canonical nuclear HUWE1 activity

Our data indicate that HAPSTR1 and HUWE1 cooperate to regulate a shared set of proteins, but that HAPSTR1 is not essential for canonical HUWE1-dependent ubiquitination. These data argue against the most parsimonious model for cofunctionality (obligate cofunctionality as a ubiquitin ligase complex), consistent with our prior experiments (Amici et al, 2022), but leave the precise biochemical nature of HAPSTR1-HUWE1 cooperation uncertain. HAPSTR1 interacts with nuclear pore proteins by mass spectrometry (Benslimane et al, 2021; Amici et al, 2022) and appears to be able to shuttle between the nucleus and the cytoplasm (Amici et al, 2022). Thus, we wondered whether HAPSTR1 functionality relates to the control of nuclear protein import/export. To test this model, we again leveraged multiplexed quantitative proteomics (Fig 2A), now to test the relative abundance of proteins in cytoplasmic versus nuclear fractions after acute HAPSTR1 depletion (16 h of siRNA treatment). We used a stable isotope labeling strategy such that HAPSTR1-depleted cells could be pooled for subcellular fractionation together with WT cells, thus controlling for any sample-to-sample technical variability in fractionation. We in addition leveraged a heavy- and light-labeled control with no genetic manipulation to rule out any changes attributable to the different medium formulations.

This experiment yielded data for 3,085 proteins. Indicating the validity of our design, the relative abundance of detected proteins in cytoplasmic versus nuclear fractions agreed strongly with independent subcellular localization annotations from the Human Protein Atlas (Thul et al, 2017) and cellular compartment gene ontology terms (Figs 2B and S2A). To determine proteins affected by acute HAPSTR1 depletion, we identified proteins significantly

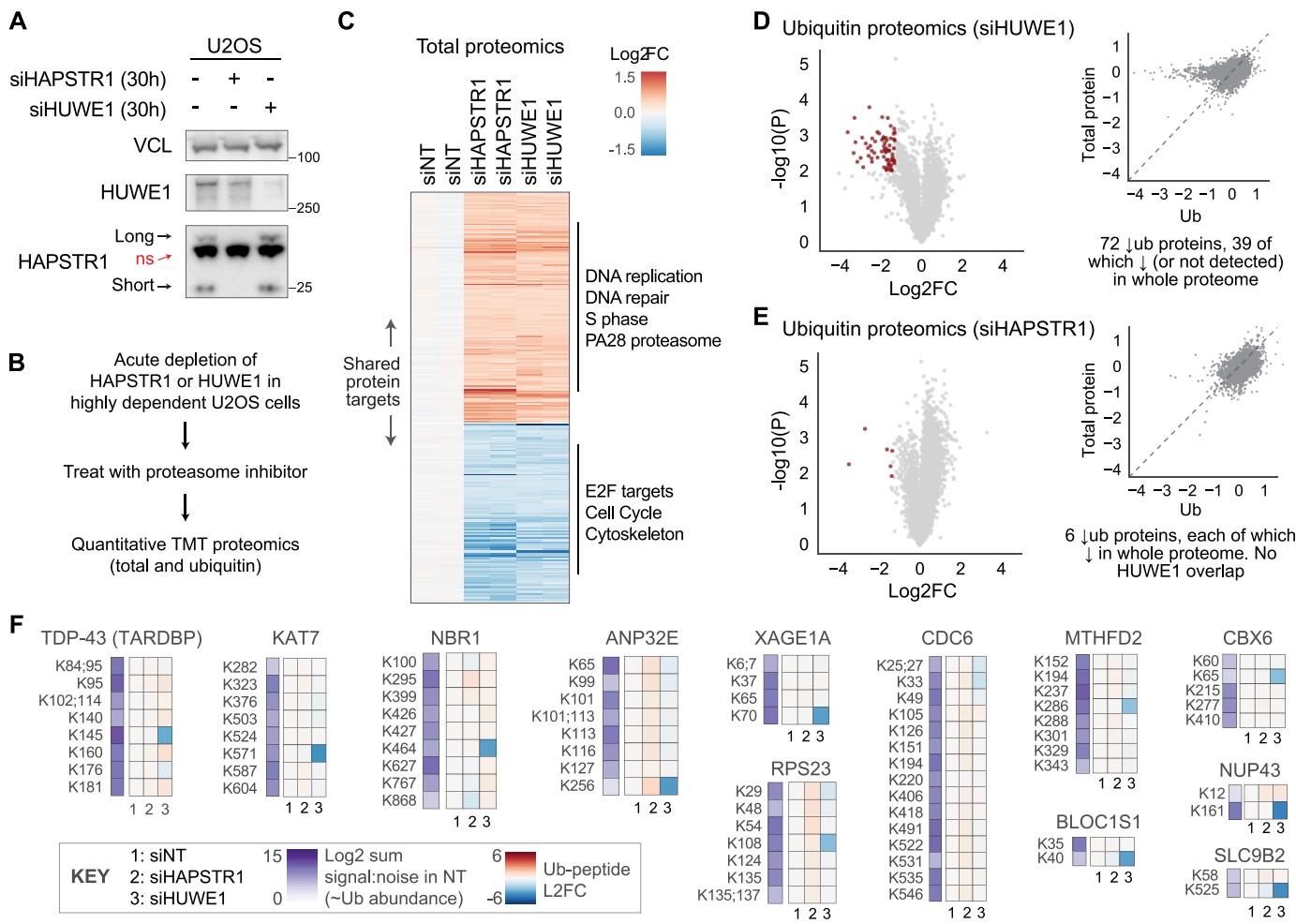

**Figure 1. Shared proteomic but not ubiquitination signatures after acute HAPSTR1 or HUWE1 depletion.**
**(A)** Representative immunoblot demonstrating that siRNA-mediated depletion of HAPSTR1 and HUWE1 is efficient within 30 h. Note that on an immunoblot, HAPSTR1 appears as a long and a short isoform with a non-specific (ns) band in between (Amici et al, 2022). **(B)** Schematic of approach. TMT, tandem mass tag. Proteasome inhibitor: MG132 50 μM × 6 h. **(C)** Proteins whose abundance was coregulated by HAPSTR1 and HUWE1 in the acute setting, with significant ontological enrichments (false discovery rate < 1 × 10$^{-5}$) noted (Subramanian et al, 2005). siNT: non-targeting control siRNA. **(D, E)** Ubiquitin proteomic data from the same cell lysates as in (C) demonstrating a substantial effect on detected ubiquitination events after loss of HUWE1, but not HAPSTR1. The inset scatterplot (upper right of each panel) compares the relative abundance of individual ubiquitinated peptides (Ub) with the abundance change of the parent protein (total protein). Log2FC versus siNT. **(F)** Example lysine-level ubiquitination data for proteins indicated as HUWE1 substrates. See Fig S1. Note key in the bottom left. L2FC, log2 fold change.

altered in at least one compartment (i.e., cytosol and/or nucleus) in the experimental (siHAPSTR1 heavy versus siNT light) but not technical (siNT heavy versus siNT light) control samples (Fig 2C). This revealed a much shorter list of proteins than our prior total proteomic experiment, consistent with the hyperacute nature of the assay. Remarkably, only one protein was significantly decreased in one compartment but increased in the other—HUWE1, which was reduced in the nuclear fraction and increased in the cytoplasmic fraction (Figs 2C and S2B).

To validate the effects of HAPSTR1 on HUWE1 localization, we depleted HAPSTR1 by siRNA in an orthogonal cell line (293T). Indeed, we found that HAPSTR1 loss resulted in nearly undetectable levels of nuclear HUWE1 (Fig 2D). Conversely, we found that the over-expression of HAPSTR1 increased HUWE1 nuclear partitioning, indicating that HUWE1 localization is sensitive to increases and decreases in HAPSTR1 abundance (Fig 2D). These findings are

consistent with a work published during the preparation of this study, which used orthogonal assays to conclude that HAPSTR1 abundance controls HUWE1 nuclear partitioning (Monda et al, 2023).

In concert with the strong genetic evidence for cooperation between HAPSTR1 and HUWE1 and our paired total/diglycine proteomic data, the observation that HAPSTR1 controls HUWE1 localization suggests a model whereby HAPSTR1 enables a nuclear function of HUWE1 not encompassed by canonical E3 ligase activity, that is, a function different from most of the cellular HUWE1 pool—which is cytoplasmic and which is not detected via diglycine proteomics. The latter may reflect a preference of HUWE1 for specific biophysical features (rather than specific, dedicated substrate proteins). For example, recent reports indicate that HUWE1 (and yeast ortholog, Tom1) extends the ubiquitin chains established by other ligases to promote clearance of damaged, aggregated, or partially ubiquitinated proteins (Kats et al, 2022;

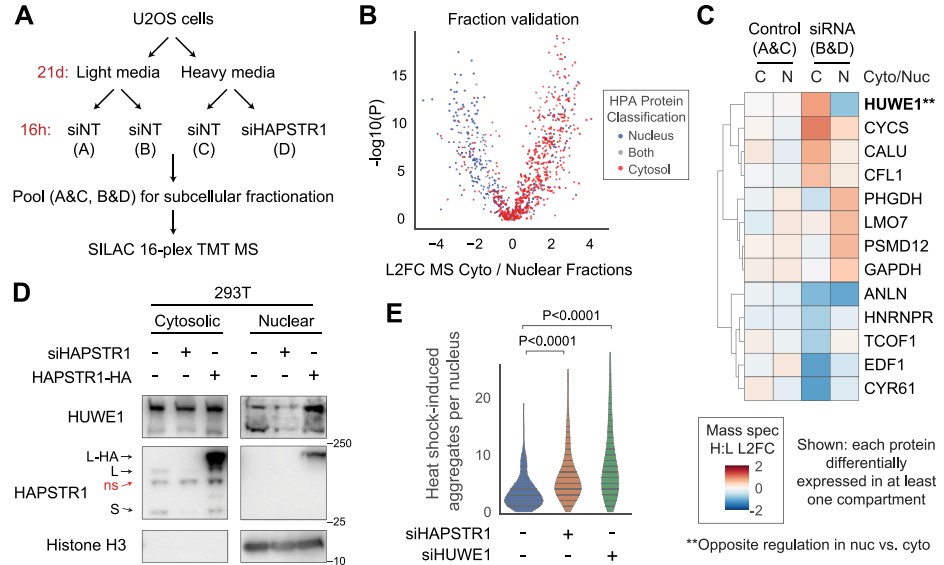

**Figure 2. HAPSTR1 enables nuclear localization of HUWE1.**
**(A)** Schematic of stable isotope labeling by amino acid in cell culture, tandem mass tag mass spectrometry (MS) experiment designed to test whether HAPSTR1 regulates protein localization. **(B)** Validation of our subcellular fractionation efficacy and MS design by comparing the relative abundance of proteins in control sample nuclear versus cytoplasmic fractions with established annotations from the Human Protein Atlas (Thul et al, 2017). **(C)** Proteins significantly differentially expressed as a function of HAPSTR1 depletion in at least one compartment with log2 fold change (L2FC) > 0.5 and $P$ < 0.01. HUWE1 was the only protein altered in different directions in both compartments. H:L indicates heavy-to-light ratio (see panel (A)). **(D)** Subcellular fractionation of an orthogonal cell line depleted of or stably overexpressing HAPSTR1. Note that the nuclear HUWE1 blot is exposed to a greater extent than the cytosolic blot (HUWE1 less abundant in the nucleus), but HAPSTR1 and histone blots are shown with the same exposure in cytosol versus nucleus. Note that on an immunoblot, HAPSTR1 appears as a long (L) and a short (S) isoform with a non-specific (ns) band in between (Amici et al, 2022). **(E)** U2OS cells were treated with siRNA for 72 h and then placed at 43°C for 2 h before immediate fixation and immunostaining for ubiquitin and DAPI (see Fig S2C). Individual nuclei were quantified for the number of aggregates using Fiji. Mann–Whitney U test, two-tailed.

Zhou et al, 2023 *Preprint*). Supporting a model wherein nuclear HUWE1 functions generally to clear partially ubiquitinated or damaged proteins, we find that cells lacking either HAPSTR1 or HUWE1 accumulate more ubiquitinated protein aggregates in the nucleus after heat stress (Figs 2E and S2C). Altogether, our data suggest that HAPSTR1 enables HUWE1 to access the nucleus and perform a non-canonical function with significant implications for cellular stress responses.

### Cellular fitness is sensitive to the abundance of HAPSTR1-HUWE1 complexes

It has been thoroughly demonstrated that HAPSTR1 loss is detrimental to cellular fitness (Meyers et al, 2017; Benslimane et al, 2021; Amici et al, 2022, 2023; Lü et al, 2022 *Preprint*; Monda et al, 2023). For example, across over 1,000 cell lines and 17,000 genes profiled in the Dependency Map project (Meyers et al, 2017; Tsherniak et al, 2017), HAPSTR1 ranks in the 88th percentile of "essentiality"—or the degree to which an individual gene's loss of function diminishes cell growth (Fig 3A). However, the phenotypic effects of promoting HUWE1 nuclear localization—via HAPSTR1 overexpression—have yet to be determined. This is of particular interest given the observation that some cancers have amplification of HAPSTR1 (Amici et al, 2022; Lü et al, 2022 *Preprint*), HAPSTR1 mRNA is induced by certain stresses (Amici et al, 2022), and HAPSTR2 increases HAPSTR1 abundance (Amici et al, 2023).

We were intrigued to find that similar to HAPSTR1 loss, HAPSTR1 overexpression had a generally detrimental effect on cellular fitness. That is, multiple cell lines overexpressing HAPSTR1 grew more slowly in culture (Fig 3B and C), even on short timescales. Observation of these growth defects by live-cell imaging suggested a defect in migration, which was further confirmed by performing a scratch assay in an orthogonal HAPSTR1-overexpressing cell line

(Fig 3D). Thus, too much and too little HAPSTR1 can diminish aspects of cellular fitness in vitro.

To better understand the consequences of HAPSTR1 overexpression, we compared the signaling consequences of HAPSTR1 overexpression with those of HUWE1 loss. We observed a striking effect: nearly every transcript altered by acute HAPSTR1 overexpression was similarly affected by HUWE1 loss (Fig 3E). HAPSTR1 overexpression in HUWE1-depleted cells did not further affect these transcripts, suggesting that HAPSTR1 overexpression affects signaling by negatively regulating HUWE1. Introduction of the HAPSTR1-F90A mutant—which does not efficiently bind HUWE1 (Amici et al, 2022)—as a control confirmed that the widespread effects of HAPSTR1 overexpression on cellular signaling are attributable to direct regulation of HUWE1 (Figs 3F and S3A–C). Together, these data suggest that overabundant HAPSTR1 has an inhibitory effect on some aspect of HUWE1 function, whether by outcompeting normal substrate(s) or through redirecting HUWE1 away from cytosolic substrates.

Finally, to test whether HUWE1 cytoplasmic function is entirely compromised upon HAPSTR1 overexpression, we leveraged DDIT4, a cytosolic protein that we have found to be the only HUWE1 substrate (other than HAPSTR1) reproducibly increased after HUWE1 loss across multiple different cell lines. We found no evidence that either overexpression or loss of HAPSTR1 affected HUWE1-mediated DDIT4 degradation (Fig 3G). Thus, although HAPSTR1 overexpression appears to result in a dominant negative effect attributable to inappropriate regulation of HUWE1, canonical HUWE1 function in the cytoplasm—to the extent it can be modeled by DDIT4 degradation—is not lost when HAPSTR1 is increased. Altogether, these data suggest that cellular fitness is sensitive to reductions in HAPSTR1, via suppression of non-canonical nuclear HUWE1 activity, as well as increases in HAPSTR1, via formation of excessive HAPSTR1-HUWE1 complexes.

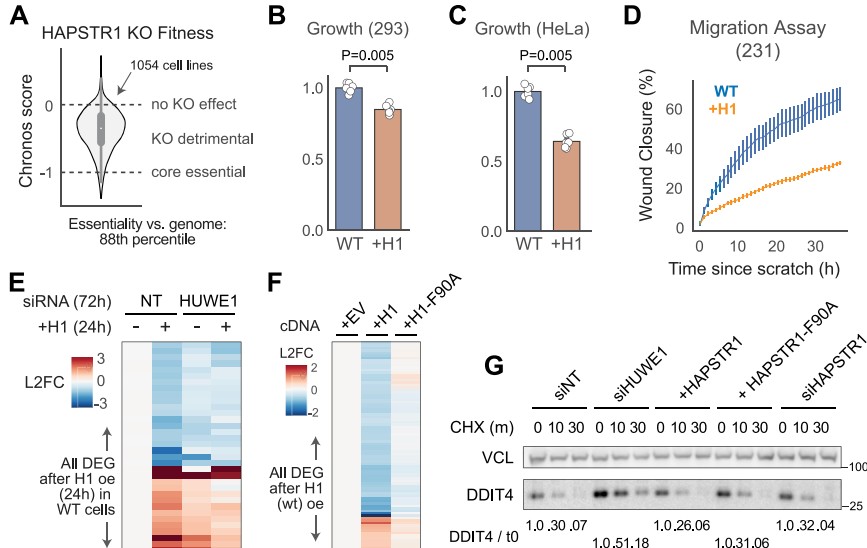

**Figure 3. Reduced cell fitness from insufficient or overabundant HAPSTR1.**
**(A)** Essentiality data from pooled CRISPR-Cas9 screens in 1,054 cell lines (Chronos score; DepMap project [Dempster et al, 2019 *Preprint*]), indicating that HAPSTR1 knockout (KO) is detrimental to cell fitness in nearly all contexts. Core essential indicates the average score of genes agreed upon to be universally required for proliferation. **(B, C)** HAPSTR1-overexpressing cells proliferate more slowly than WT HEK293T (293) or HeLa cells over 2 d. Mann–Whitney *U* test. **(D)** HAPSTR1 overexpression produces a migration defect, as defined by defective closure of a scratch wound in serum-starved media. N = 5, MDA-MB-231 cells. **(E)** Transient (24 h) overexpression of HAPSTR1 in 293T cells produces gene expression changes, which mimic HUWE1 loss and are suppressed by HUWE1 loss. DEG, differentially expressed genes; NT, non-targeting. Data from Amici et al (2022). **(F)** Mutating the HUWE1-binding site on HAPSTR1 (F90A) diminishes the signaling consequences of HAPSTR1 overexpression in 293T cells. Transient transfection, 48 h. EV, empty vector. **(G)** Neither HAPSTR1 overexpression nor knockdown impacts the degradation kinetics of a dedicated HUWE1 substrate, DDIT4.

## Regulation of HAPSTR1 stability by a coessential ubiquitination network

Given the importance of maintaining tight control of HAPSTR1 abundance, we sought to identify factors that mediate titration of HAPSTR1 abundance. We reasoned that such genes, if critical for optimal pathway function, would be phenotypically "coessential" with HAPSTR1. That is, their loss-of-function effects would mimic those of HAPSTR1, as we previously observed for HUWE1 (the top coessential gene for HAPSTR1) (Amici et al, 2022) and as have been widely observed for genes that cooperate in the same pathway (Pan et al, 2018; Kim et al, 2019; Amici et al, 2021). To this end, we applied our coessentiality tool, FIREWORKS (Amici et al, 2021), to CRISPR-Cas9 fitness screening datasets encompassing hundreds of diverse cancer cell lines (Tsherniak et al, 2017) or to a single cell line (RPE1-hTERT, p53-null) cultured in the presence of various genotoxic stressors (Olivieri et al, 2020).

These analyses identified several genes as potentially cofunctional with HAPSTR1 (Fig 4A). Two candidate genes stood out because of strong coessentiality and prior connections to HUWE1 and/or HAPSTR1 (Fig 4B). These genes encode the deubiquitinase USP7 (#2 HAPSTR1 correlate in DepMap screens, behind only HUWE1) and the ubiquitin ligase TRIP12 (#1 correlate in DNA damaging agent screens, #10 in DepMap screens). Multiple functional interactions have been reported between the physically interacting proteins USP7 and HUWE1 (Sowa et al, 2009; Khoronenkova & Dianov, 2013; Thompson et al, 2014; Wu et al, 2016), and USP7 itself interacts with HAPSTR1 (Amici et al, 2022, 2023). Furthermore, the E3 ligase TRIP12 shares homology with HUWE1 and has been proposed to cooperate with HUWE1 in degradation of a model ubiquitination substrate (Poulsen et al, 2012).

We first assessed these genes by depleting them in parallel via siRNA (Figs 4C and S4). As predicted based on the coessentiality data, loss of any of the HAPSTR1-coessential genes resulted in the increased expression of p21, a reliable marker of HAPSTR1 pathway loss in p53-WT cells (Amici et al, 2022, 2023; Lü et al, 2022 *Preprint*) (Fig 4C).

Intriguingly, despite broadly sharing a functional role in the cell, we found that—like HUWE1—TRIP12 promoted HAPSTR1 degradation, whereas USP7 promoted HAPSTR1 stability (Fig 4D). The function of TRIP12 in mediating HAPSTR1 degradation was achieved in cooperation with HUWE1, with loss of either factor sufficient to augment HAPSTR1 levels but loss of both conferring the greatest stability of HAPSTR1 (Fig 4D). We note that the effect of USP7 on HAPSTR1 steady-state abundance was small, but that stability was strongly affected. Fittingly, the overexpression of USP7 promoted a striking accumulation of HAPSTR1 (Fig 4E).

Altogether, these data indicate that HAPSTR1 abundance regulates a non-canonical nuclear HUWE1 activity and that a network of coessential enzymes that regulate ubiquitination (HUWE1, TRIP12, and USP7) interacts to titrate HAPSTR1 stability and prevent pathway misregulation.

## Essential role of the HAPSTR1 pathway in humans and mice

With an improved understanding of the molecular aspects of the HAPSTR1-HUWE1 pathway, we sought to better understand the role of this pathway in mammalian physiology. We first leveraged human genetics to assess the importance of HAPSTR1 for human evolutionary fitness. Mutational analysis using the gnomAD database (Karczewski et al, 2020) suggested that HAPSTR1 loss of function is intolerable in humans (Fig 5A). That is, mutations predicted to cause HAPSTR1 loss of function are exquisitely rare (observed/expected ratio, 0.06; 95% CI, 0.02–0.29), yielding a predicted loss-of-function intolerance (pLI) score of 0.98 (Fig 5A). Given that 0.9 is a widely used cutoff for haploinsufficient genes (Karczewski et al, 2020), these data strongly suggest that humans require two functional copies of HAPSTR1.

To better understand HAPSTR1's role in organismal physiology, we next generated conditional knockout Hapstr1 (1810013L24Rik)

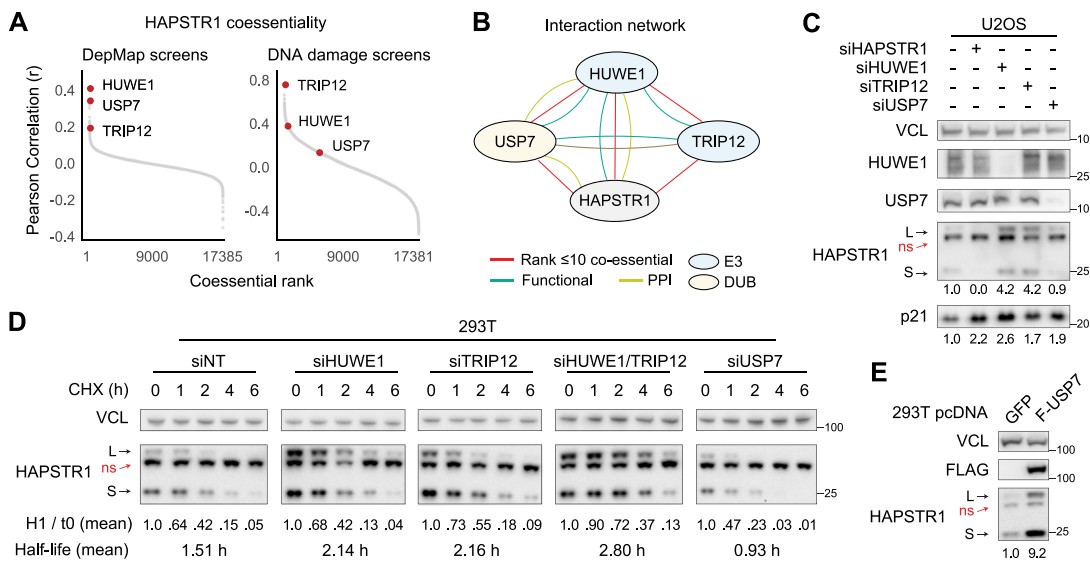

**Figure 4. Coessential ubiquitination network titrates HAPSTR1.**
**(A)** Similar to HUWE1, USP7 (#2 DepMap correlate) and TRIP12 (#1 DNA damage screen correlate, #10 DepMap correlate) phenocopy HAPSTR1 in unbiased phenotypic screening datasets. **(B)** Schematic of known relationships between the indicated genes. PPI, protein–protein interaction; DUB, deubiquitinase; E3: ubiquitin ligase. Functional indicates a reported regulatory or collaborative action. **(C)** Knockdown of individual factors in the HAPSTR1 coessentiality network results in similar regulation of a downstream pathway target (p21) but variable effects on HAPSTR1 abundance. Note that on an immunoblot, HAPSTR1 appears as a long (L) and a short (S) isoform with a non-specific (ns) band in between (Amici et al, 2022). TRIP12 immunoblots were non-specific; see Fig S4 for additional siRNA validation. **(D)** Protein stability assays for HAPSTR1 demonstrate that TRIP12 cooperates with and is partially redundant with HUWE1 in the degradation of HAPSTR1, whereas USP7 promotes HAPSTR1 stability. Quantification is relative to assay starting quantity for the indicated genotype; that is, siHUWE1 and siTRIP12 start with more absolute HAPSTR1, but degradation is represented as a percentage of initial 40 µg/ml cycloheximide. **(E)** Transient overexpression of FLAG (F)-tagged USP7 results in profound HAPSTR1 stabilization over 2 d. Quantified Western blots represent at least three independent experiments.

mice. LoxP sites were knocked in flanking *Hapstr1* exons 1 and 2 such that after Cre-mediated recombination, both canonical Hapstr1 transcripts would be lost and the reading frame for any alternatively initiated transcripts would be frameshifted (Fig 5B). *Hapstr1*-floxed mice were then bred with mice carrying a Cre, which is active in the early embryo (Sox2Cre), creating a whole-body knockout at the embryo stage, or which could be temporally controlled by the administration of tamoxifen (Cre-ERT2). We note that Hapstr1 is expressed ubiquitously across organs in mice (Amici et al, 2022, 2023).

We quickly observed that attempts to generate complete Hapstr1 knockout animals via embryonic recombination were futile. That is, we failed to identify a single viable Hapstr1-null mouse across 68 litters, indicating that Hapstr1 is required for life in mice (Fig 5C). On the contrary, heterozygous knockouts were born at approximately the expected rate and did not have any obvious phenotypic defects (Figs 5C and S5A–C).

To further investigate the lethality of homozygous Hapstr1 loss, we analyzed embryos from timed breeding pairs expected to yield WT, heterozygous, and knockout animals. Hapstr1-null embryos with no obvious morphological defects (frank color changes, size differences, or malformations) were readily identified at embryonic day 11.5 (E11.5), indicating no requirement for Hapstr1 in basic embryogenesis (Fig 5D). On the contrary, by E14.5, null mice were rarer and easily distinguished from littermates because of small size, and we did not identify any null mice at E18.5 (Fig 5D and E). Altogether, these data indicate that Hapstr1 is essential for life in mice, with knockout typically resulting in death in utero around E15-18.

## Growth and signaling defects in Hapstr1-null primary cells

To better understand the defects leading to embryonic lethality of Hapstr1-null mice, we generated MEFs from E11.5 WT and KO embryos. Consistent with the role of human HAPSTR1 in controlling HUWE1 localization, we found that primary cells from Hapstr1-null embryos had nearly undetectable levels of nuclear HUWE1, but increased abundance of cytosolic HUWE1 (Fig 6A). Next assessing cellular fitness, we found that Hapstr1-null MEFs proliferated analogously to WT cells after the initial plating (Fig 6B). However, Hapstr1-null MEFs quickly faltered, resulting in a 22-fold overall reduction in replicative potential (Fig 6B).

To broadly assess the signaling consequences, which lead to diminished fitness of KO primary cells, we performed RNA-sequencing on three independently derived KO and WT MEF lines at passage 4 (p4). As in human cancer cells (Amici et al, 2022), Hapstr1 loss provoked a pleiotropic set of gene expression changes (Fig 6C). Most obvious among the changes attributable to Hapstr1 loss was suppression of TGF-β signaling. That is, genes known to be induced by TGFB1 treatment in MEFs were down-regulated in Hapstr1 knockouts, whereas genes suppressed by TGFB1 treatment were up-regulated (Fig 6C). Other differential pathways in KO MEFs included activation of oxidative and abiotic stress responses, as well as suppression of an epithelial-to-mesenchymal transition. Thus, primary cells from Hapstr1-null embryos have greatly diminished fitness in the culture associated with HUWE1 mislocalization and widespread signaling alterations.

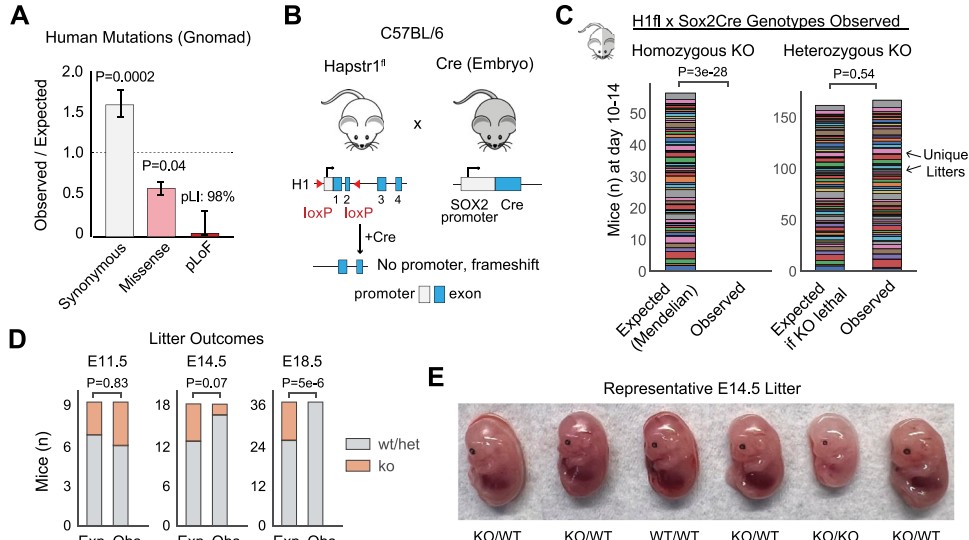

**Figure 5. HAPSTR1 is essential in humans and mice.**
**(A)** Analysis of human HAPSTR1 mutation frequency using data and parameters from gnomAD (Karczewski et al, 2020). pLOF, predicted loss-of-function mutations. *P*-values are derived from z-scores versus the rest of genome. pLI: computed probability of loss-of-function intolerance (haploinsufficiency). Note that 90% pLI is used clinically to call haploinsufficient genes. **(B)** Schematic of the relevant mouse strains created and used in this article. **(C)** Number of observed versus expected (by Mendelian inheritance) homozygous and heterozygous knockout (KO) Hapstr1 mice. Two-tailed binomial proportion test. **(D)** Observed (o) versus expected (e) numbers of KO Hapstr1 mice, as in (C), at embryonic day 11.5, 14.5, or 18.5. Two-tailed binomial proportion test. **(E)** Representative litter at E14.5 demonstrating six WT or heterozygous mice and one phenotypically dissimilar KO.

## Hapstr1 is essential for life even in p53-null mice

A core phenotype of HAPSTR1-HUWE1 pathway loss is activation of the growth-suppressive transcription factor p53 (Benslimane et al, 2021; Amici et al, 2022, 2023; Lü et al, 2022 *Preprint*; Monda et al, 2023). Consistent with this observation, we found increased p53 levels in Hapstr1-null MEFs compared with controls (Fig 6D). However, given the pleiotropic signaling effects of HAPSTR1 loss, it has remained unclear to what extent p53 activation mediates Hapstr1 phenotypes.

We first assessed the effect of p53 knockout on the growth of Hapstr1-null MEFs. P53 knockout via CRISPR-Cas9 substantially increased growth in Hapstr1-WT and Hapstr1-KO MEFs (Fig 6E). Comparing the relative growth effects of Hapstr1 loss in p53-WT versus p53-KO MEFs, we observed a partial, but incomplete rescue of the growth defect attributable to Hapstr1 loss (Fig 6E). This is consistent with data from the DepMap Project, which indicate that HAPSTR1 is most essential for growth in cell lines with intact p53 signaling, but still important for cells that lack p53 (Fig 6F).

Finally, to assess whether p53 mediates the developmental lethality of Hapstr1-null mice, we crossed germline p53 knockout mice with our conditional germline Hapstr1 animals. Even among mice completely lacking p53, no Hapstr1 nulls were identified (Fig 6G). Thus, although p53 activation is a highly consistent consequence of HAPSTR1-HUWE1 pathway impairment, Hapstr1 is required for cellular and organismal fitness even in the absence of p53.

## Induced HAPSTR1 knockout in adult mice reduces fitness

Our data indicate that Hapstr1 must be present during development for mice to pass the perinatal stage. However, whether Hapstr1 is required in fully developed adult mice is unknown. We thus induced HAPSTR1 knockout via tamoxifen administration (intraperitoneal injection and supplemented chow) in 3-mo-old adult *Hapstr1^fl/fl CreERT2^+* mice (Fig 7A). PCR indicated that the intended recombination event occurred (Fig 7B), albeit with incomplete penetrance, which may in part relate to outcompetition of

knockout cells by WT neighbors that did not recombine (as we have observed with population HAPSTR1 knockout experiments in cell culture). Only the liver demonstrated poor recombination overall, consistent with a known limitation of this inducible Cre model (Hayashi & McMahon, 2002). We did not observe any evidence of knockout absent tamoxifen. Immunoblots further confirmed partial knockout of HAPSTR1 at the tissue level (Fig S6A).

Despite incomplete knockout, we found that HAPSTR1 induced knockout (iKO) resulted in robust weight loss over the span of 6 wk (Fig 7C). This effect was observed in mice of both sexes, but more pronounced males. Indeed, male iKOs lost at least 15% of their bodyweight within 3 wk, with one mouse becoming so deconditioned as to require dietary rescue intervention (Fig S6B).

To investigate whether the observed weight loss was attributable to substantial damage to a specific organ, we performed histology on organs representing a broad survey of morphological features and tissue types (brain, spleen, kidney, lung, heart, testis, and colon). We found that the overall organization and morphology of organs were preserved, suggesting that acute, albeit partial, knockout of HAPSTR1 in adult mammals does not acutely result in specific organ failure. The only notable distinction between WT and iKO organs was observed in the colon, where an increased number of inflammatory foci were observed (Fig 7D). Morphologically, these included larger lymphoid aggregates and smaller patches of infiltrating mononuclear cells, potentially relating to ongoing cell death or inflammatory signaling processes in the rapidly dividing cells of the colon. Gastrointestinal tract distress may also explain the observed weight loss and, in the case of two of three iKO males, weight regain late in the experiment attributable to massive swelling and distension of the cecum (Fig S6B and C).

Finally, because Hapstr1 has previously been linked to processes relevant to wound repair, such as migration (Fig 3D and [Amici et al, 2022]) and paracrine/chemokine signaling (Fig 6C and [Amici et al, 2022; Monda et al, 2023]), we sought to test whether Hapstr1 loss reduces adaptability after wound-like stress. Chemical hair removal (depilation) induces a similar response to wound repair and can be

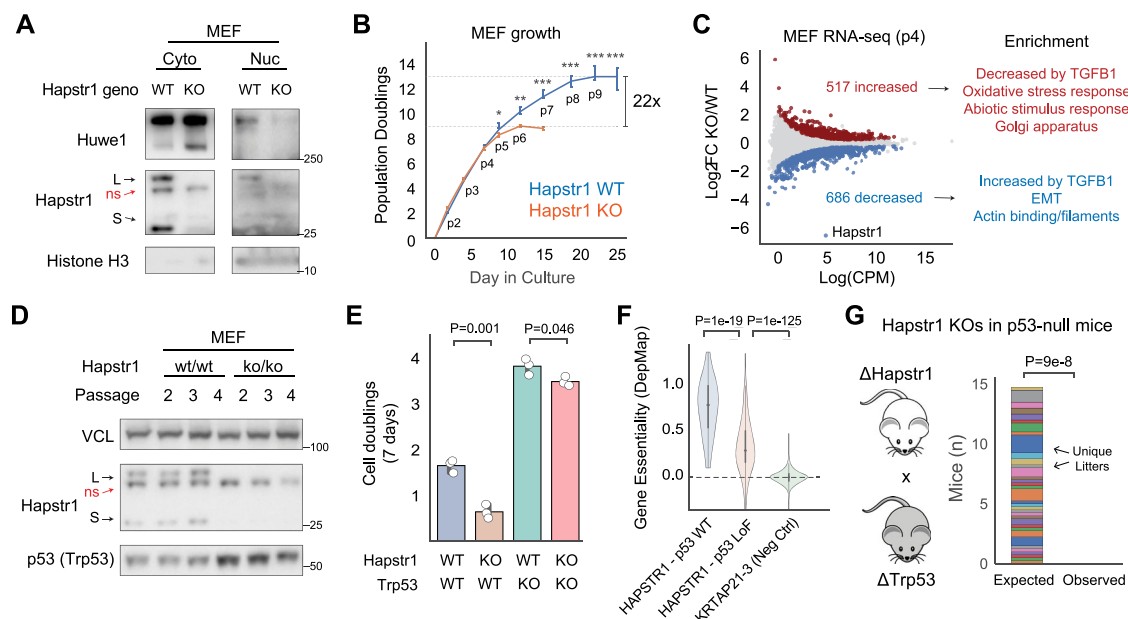

**Figure 6. Widespread defects in Hapstr1-null primary cells and lack of rescue with p53 codeletion.**
**(A)** Immunoblots of Hapstr1-null MEFs reveal Huwe1 cytoplasmic mislocalization. **(B, D)** Early growth arrest of Hapstr1-null MEFs derived from embryos in (D). N = 3 independent mouse-derived MEF lines per group. Passage numbers indicated below lines. Two-tailed t test, ***$P < 0.001$, **$P < 0.01$, *$P <0.05$. **(C)** RNA-seq of Hapstr1 KO versus WT MEFs at passage 4 (before stark growth changes) and significant gene set enrichments among the significantly altered transcripts. N = 3 independent mouse-derived MEF lines per group. **(D)** Increased p53 abundance in Hapstr1-null MEFs. **(E)** P53 (Trp53) KO by CRISPR-Cas9 in existing MEF lines incompletely rescues the growth defects attributable to Hapstr1 loss. **(F)** Essentiality scores (Chronos * –1; where 1 represents a core essential gene, and 0 represents no fitness effect) of HAPSTR1 across human cancer cell lines with either intact p53 growth advantage after deletion (essentiality score ≤ –1) or p53 loss of function (essentiality score ≥ 0). Two-tailed t test. A non-expressed control gene is used to represent a null effect (KRTAP21-3). **(G)** Among mice completely lacking p53, the number of observed versus expected (by Mendelian inheritance) homozygous KO Hapstr1 mice, as in Fig 5C. Two-tailed binomial proportion test.

tested in mice in a non-invasive fashion (Arwert et al, 2012; Rodgers et al, 2017). We thus applied Nair to hindlimbs of WT and iKO mice 2 wk before scheduled euthanasia. Whereas WT mice regrew essentially all lost hair, HAPSTR1-null mice had notably less regrowth, consistent with the hypothesis that HAPSTR1 augments reparative processes at the tissue level (Fig 7E).

Altogether, our data indicate that the conserved and tightly regulated HAPSTR1 protein enables a non-canonical nuclear activity of HUWE1 essential for mammalian life.

# Discussion

In the past few years, HAPSTR1 (formerly: C16orf72) has gone from entirely unstudied to a gene of widespread interest because of strong phenotypes in unbiased, high-throughput screens. Initial reasons to perform targeted investigations of HAPSTR1 have included its proresilience effects in the contexts of cancer-associated stresses (Amici et al, 2022), replication stress (Hustedt et al, 2019), or telomerase inhibition (Benslimane et al, 2021); its striking role in the suppression of p53 (Lü et al, 2022 Preprint); its intriguing stress-associated transcriptional regulation (Chakraborty et al, 2022); and its uniquely robust stabilization after HUWE1 loss (Monda et al, 2023). Clear across these studies are that HAPSTR1 and HUWE1 represent central components of a pathway critical for resilience and stress response signaling processes.

One key question that has proven difficult to answer is the precise biochemical nature of HAPSTR1-HUWE1 cooperation. The simplest model would be one wherein HAPSTR1 is a cofactor for HUWE1-mediated ubiquitination of specific substrates. Here, we coupled gold-standard proteomic approaches with the acute depletion of HUWE1 to identify dozens of putative substrate proteins for HUWE1. Independent of HAPSTR1, this dataset may serve as a rich resource for the community of researchers interested in the multifaceted and highly disease-relevant HUWE1 protein.

With regard to HAPSTR1, on the contrary, we were surprised to find that none of the HUWE1-dependent ubiquitination events identified in our experiment required HAPSTR1. This finding was particularly intriguing given a shared effect on the abundance of many proteins in the same cell lysates—as well as the multitude of other proteomic, transcriptomic, and functional datasets, which show similar outcomes when either HAPSTR1 or HUWE1 is lost (Olivieri et al, 2020; Amici et al, 2022; Monda et al, 2023). We propose that this divergence represents the separation of molecular function between HAPSTR1 and HUWE1. That is, consistent with our prior targeted experiments investigating HUWE1 substrates DDIT4 and MCL1 (Amici et al, 2022), HAPSTR1 is not required for the canonical activity of the largely cytoplasmic (Xu et al, 2016; Monda et al, 2023) HUWE1 ubiquitin ligase. Instead, data here and in a recent article (Monda et al, 2023) now suggest that HAPSTR1 mediates a nuclear subset function of HUWE1. While representing only a subset of HUWE1's functionality, this nuclear activity is clearly of major physiological importance, given how much of the loss-of-

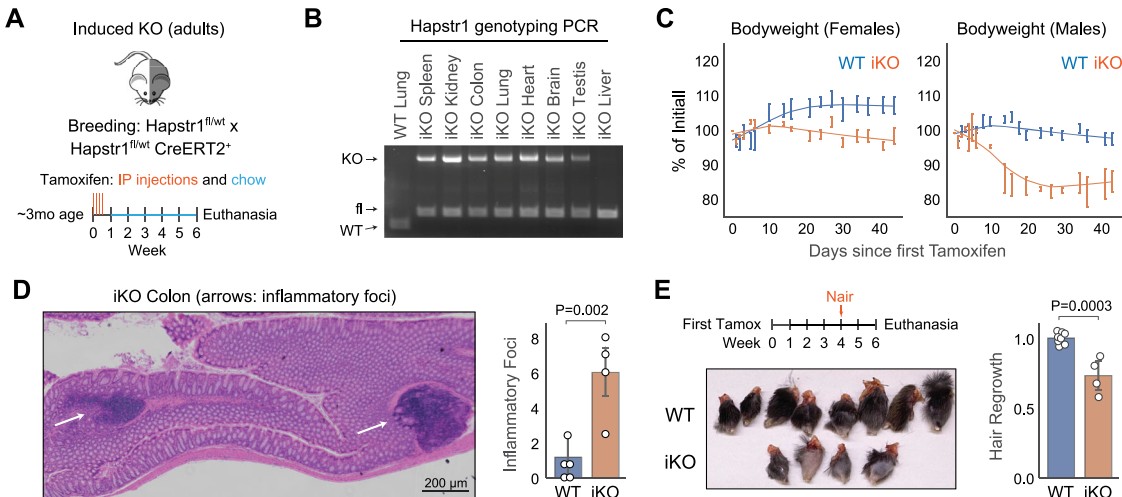

**Figure 7. Effects of induced Hapstr1 knockout in fully developed adult mice.**
**(A)** Schematic of the experiment for induced knockout (iKO) of Hapstr1 in mice, beginning at ~3 mo of age. IP, intraperitoneal; fl, floxed (flanked by loxP); wt, wild type. **(B)** Representative PCR using three primers, creating bands of different sizes for WT, floxed, and recombined/iKO Hapstr1, demonstrates mosaic knockout across multiple organs. **(C)** Hapstr1 iKO reduces bodyweight in male and female mice. Three to five mice per group, with littermate controls who received tamoxifen as controls. See Fig S6. **(D)** Example inflammatory foci found in an iKO colon section and the frequency of these foci in all tested colons from male and female mice. Two-sample *t* test. **(E)** Schematic of the experiment to chemically remove hair before scheduled euthanasia; photograph of the legs at the time of harvest; and quantification of hair regrowth (relative darkness, by grayscale pixel density). Two-sample *t* test.

function phenotypes attributable to HUWE1 in proliferating cells can be explained by loss of HAPSTR1 alone (Amici et al, 2022). It is worth noting, however, that many canonical HUWE1 functions are control lineage and developmental signaling events (Kao et al, 2018), potentially explaining the earlier developmental lethality (~E12-14) of HUWE1-null mice (Hao et al, 2012; Kon et al, 2012) versus the later developmental lethality (~E15-18) of HAPSTR1-null mice.

The natural model following the findings that HAPSTR1 directs HUWE1 to the nucleus is that nuclear HUWE1 then acts to ubiquitinate nucleus-restricted substrates. Yet—given our observation that HAPSTR1 is not required for HUWE1's effects on the diglycine proteome—which substrates? We postulate two non-mutually exclusive possibilities for a non-canonical nuclear HUWE1 activity, which could explain this finding.

The first possibility is a "modification" model, wherein nuclear HUWE1 ubiquitinates proteins with a specific biophysical feature or modification, rather than serving as a dedicated ligase for specific nuclear protein(s) in their normal state. These could be, for example, orphaned subunits of protein complexes, which now expose hydrophobic residues, as previously suggested for HUWE1 in the cytosol (Sung et al, 2016; Xu et al, 2016). These modified proteins would represent a small portion of the overall cellular pool of the indicated protein and thus be difficult to detect in proteomic experiments. The second non-exclusive possibility is an E4 ligase model, wherein HUWE1 binds mono- or oligo-ubiquitinated proteins and elongates or diversifies the existing ubiquitin chain. An analogous activity, which cannot be detected from standard diglycine proteomic experiments, has now been demonstrated for human and yeast HUWE1/Tom1 (Kats et al, 2022; Zhou et al, 2023 Preprint). Both models of this non-canonical nuclear function of HUWE1 are consistent with the difficulty in identifying dedicated substrates of the HUWE1 ligase, which is essential and pleiotropic in

nearly all cells but affects vastly different proteins depending on cell type and context (Thompson et al, 2014; Amici et al, 2022; Monda et al, 2023). These models are further consistent with the multiple connections of this pathway to stress, which is expected to increase the abundance of damaged, atypical, and/or partially ubiquitinated proteins in the nucleus.

Hints that HAPSTR1's activity is tightly regulated came from the observation that, like HAPSTR1 loss, HAPSTR1 overexpression detrimentally affects certain aspects of cellular fitness. HAPSTR1 overexpression provoked signaling changes similar to HUWE1 loss, which were nullified by introducing a mutation that abolishes HAPSTR1-HUWE1 complex formation. Thus, cells appear to prefer a specific quantity of HAPSTR1-HUWE1 complexes, with too many inhibiting aspects of HUWE1 function, and too little preventing HUWE1 function in the nucleus. Consistent with this model, we found that ubiquitin ligase TRIP12 and deubiquitinase USP7—which oppose each other in titrating HAPSTR1 stability—are both coessential for optimal pathway function and cellular fitness. We postulate that the "optimal" quantity of HAPSTR1 in the cell is context-dependent. For example, HAPSTR2 robustly stabilizes HAPSTR1, but is only present in neural and germline tissues (Amici et al, 2023), suggesting that these tissues may benefit from a slight bias toward increased nuclear HUWE1 activity. Similarly, the modest overexpression of HAPSTR1 in tumors (Amici et al, 2022; Lü et al, 2022 Preprint) or ectopic HAPSTR2 overexpression (Amici et al, 2023) can promote p53 degradation, which may be useful in contexts where p53 is acutely growth-restrictive. Altogether, our data suggest a model whereby HAPSTR1 abundance dictates the landscape of HUWE1 function and that multiple distinct inputs feed into the control of HAPSTR1 activity.

Beyond titrating HAPSTR1, a direct functional role of TRIP12 in the pathway is also intriguing to consider. That is, TRIP12 is a nuclear

protein (Thul et al, 2017; Larrieu et al, 2020) and has been proposed to act as an E3 to HUWE1's E4 in the degradation of a model ubiquitin fusion protein (Poulsen et al, 2012). Our identification of HAPSTR1 as the first physiological cosubstrate for these two coessential HECT ubiquitin ligases raises the possibility that TRIP12 cooperates more broadly with HUWE1 in nuclear substrate degradation.

Finally, the creation of conditional Hapstr1 mice allows the first insights into the pathway's significance for physiology in living animals. Human genetics suggested an essential role of HAPSTR1 in mammals, a prediction confirmed by the developmental lethality of germline Hapstr1-null mice. Intriguingly, Hapstr1 was not required for basic embryogenesis, indicating that a great deal of physiological development and organogenesis can proceed absent Hapstr1. Experiments in explanted fibroblasts from Hapstr1-null embryos confirmed that Hapstr1 loss abrogated Huwe1 nuclear access, resulting in widespread signaling defects and impaired cellular fitness. Among the observed signaling defects was a failure to suppress p53, the famous tumor suppressor that has garnered interest as the possible primary effector of the HAPSTR1 pathway. However, unlike in the case of the dedicated p53 regulator Mdm2 (Jones et al, 1995), we find that Hapstr1 loss cannot be rescued by codeletion of p53. This, alongside the observation that there are yeast orthologs for HAPSTR1 and HUWE1 but not p53, argues against p53 suppression as the primary function of the Hapstr1–Huwe1 pathway.

In conclusion, our multidisciplinary investigation provides insight into the function, regulation, and physiological significance of a conserved and disease-relevant pathway by which cells combat the stresses of life.

# Materials and Methods

### Statistics and data analysis

All data analysis was performed using standard modules in Python (v3.7.6) as follows. Bar, box, line, strip (individual point), and violin plots were created using the respective functions in Seaborn (v0.11.1) and Matplotlib (v3.5.2). Data cleaning and statistical analyses used standard functions in Pandas (v1.1.3), NumPy (v1.21.1), SciPy (v1.6.2), and Statannot (v.0.2.3). Individual replicates highlighted in figures and figure legends refer to biological replicates (independent samples/experiments) rather than technical replicates. Where technical replicates were performed, the values were averaged to a single biological replicate value.

### Cell lines

Standard cell lines were obtained from the ATCC. MEFs were generated using a standard approach. Briefly, a timed breeding cage was established to determine the date of pregnancy. After 11 d, fetuses were isolated and washed in PBS, the internal organs and heads removed, and then digested in 0.05% trypsin for 3 h before neutralization in FBS and plating for cell culture. Isolated fibroblasts were then cultured in high-glucose DMEM (Corning)

supplemented with 10% FBS and 1% antibacterial/antimycotic (Gibco). Cells were passaged with trypsin and tested regularly for Mycoplasma contamination.

### Transfections and lentiviral infections

SMART pool siRNAs (Horizon/Dharmacon) were obtained for each target gene of interest, as well as a non-targeting sequence, and reverse-transfected using RNAiMAX (Thermo Fisher Scientific) using the manufacturer's recommended reagent ratio. Cells were harvested 72 h after siRNA transfection by default. For acute knockdown experiments, cells were plated overnight and then forward-transfected the next morning. Plasmid transfections used Lipofectamine 3000 (Thermo Fisher Scientific) and followed the manufacturer's protocols. For lentivirus, lentiviral vectors containing DNA constructs of interest were reverse-cotransfected with pMD2.G and psPAX2 into 293T cells using Lipofectamine 3000 at the ratio recommended by the manufacturer. Media on transfected cells were changed 16 h post-transfection, and then, the new virus-containing media were harvested 48 h after initial transfection. Media were then centrifuged at 1,000$g$ and filtered with an 0.45-$\mu$m filter to yield packaged lentivirus. Lentivirus was then added directly to cells for 24 h for transduction. Selection was achieved using 2 $\mu$g/ml puromycin, 10 $\mu$g/ml blasticidin, or 250–500 $\mu$g/ml hygromycin. An empty vector was used as a control for stable cell line experiments.

### Cell lysis and immunoblots

Unless otherwise specified, cells were lysed in buffer composed of 50 mM Tris, pH 7.5, 100 mM NaCl, 1% Triton, 0.2 mM EDTA, 5% glycerol, and 1 mM PSMF. Cell lysis was achieved by vortexing on ice followed by sonication using Bioruptor (Diagenode) for 10 cycles of 30 s on, 30 s off. Lysates were cleared by centrifugation for 10–15 min at 21,000$g$ and 4°C. Protein concentration was assessed using a BCA assay (Thermo Fisher Scientific) and then equalized across samples, typically to a final concentration of 1–2 mg/ml. NuPAGE 4–12% Bis–Tris gradient gels were used for all immunoblots (Thermo Fisher Scientific). Imaging was performed with Bio-Rad ChemiDoc Touch Imaging System (732BR0783) after incubation in HRP substrate (Immobilon, Millipore). Blots were analyzed using ImageLab v6.0.1 (Bio-Rad). The antibodies used were as follows (name, catalog number, dilution): HAPSTRl (OTl2B8; 1:1,000; OriGene), FLAG (F3165; 1:5,000; Sigma-Aldrich), HA (26183; 1:5,000; Thermo Fisher Scientific), HUWE1 (ab70161; 1:1,000; Abcam), vinculin (V9131; 1:10,000; Sigma-Aldrich), HO-1/HMOXl (NBPl-97507; 1:1,000; Novus), p21/CDKN1A (2947; 1:1,000; CST), p53/TP53 (P6749; 1:2,000; Sigma-Aldrich), HRP anti-rabbit IgG secondary (7074; 1:10,000; CSF), HRP anti-mouse IgG (31430; 1:10,000; Thermo Fisher Scientific).

### Subcellular fractionation

All centrifuge steps were at 4°C. Cells and tubes are all kept on ice throughout the protocol. Buffer A consisted of 10 mM Hepes (pH 7.9), 1.5 mM MgCl$_2$, 10 mM KCl, and protease inhibitors (added just before use). Buffer B consisted of 20 mM Hepes (pH 7.9), 1.5 mM MgCl$_2$,

420 mM NaCl, 25% (vol/vol) glycerol, 0.2 mM EDTA, and protease inhibitors (added just before use).

Cells were washed once in cold PBS and then harvested by scraping in cold PBS, then spun at 400*g* × 4 min. The supernatant was removed by aspiration. Cells were resuspended in buffer A at ~400 ml per 10 × 10$^6$ cells (more or less depending on the protein concentration of intended cells). NP-40 was then added to the cells in buffer A (to a final concentration of 0.5–1%). Cells were gently vortexed (brief pulses not at full speed) to 5–10 s at a time and then put back on ice for 2 min, repeated five times. Nuclei were pelleted at 400*g* × 5 min. The supernatant (crude cytosol) was placed in a new tube. Nuclei were then washed in buffer A, fully resuspended, gently vortexed a few times, and then spun again at 400*g* × 4 min. The supernatant was then removed from the washed nuclei. Washed nuclei were then resuspended in buffer B and vortexed aggressively on and off ice for 15 min. During this portion, crude cytosol was spun at 12,000*g* × 10 min. The supernatant is the cytosol fraction. Finally, lysed nuclei were sonicated using Bioruptor (30 s on, 30 s off, 15 cycles). Lysed nuclei were then spun at 12,000*g* × 10 min, and the supernatant was retained as a nuclear fraction.

## Total proteome and diglycine mass spectrometry

U2OS cells were grown in 15-cm plates, forward-transfected with the indicated siRNA as described above, and then allowed to grow for 30 h before the addition of 25 *μM* MG132 for 6 h. Cells were then washed briefly in cold PBS and scraped in a cold room, followed by pelleting at 4°C for 5 min at 1,000*g*.

Lysates for mass spectrometry were then prepared essentially as previously described (Navarrete-Perea et al, 2018; Li et al, 2021). Cells were lysed in a buffer containing 200 mM EPPS, pH 8.5, 8 M urea, 20 mM N-ethylmaleimide, 1x Roche complete protease inhibitor cocktail, and 1 mM PMSF. Lysis and protein concentration assessment were performed as described above. After lysis, each sample was reduced with 5 mM TCEP. Cysteine residues were alkylated using 10 mM iodoacetamide for 20 min at RT in the dark. Excess iodoacetamide was quenched with 10 mM DTT. 2.5 mg of each proteome was precipitated and resolubilized in 200 mM EPPS, pH 8.5. Samples were digested with Lys-C (1:50) overnight at RT and subsequently with trypsin (1:100) for 6 h at 37°C. Digested peptides were desalted using Sep-Pak.

Peptides harboring a di-Gly remnant were enriched using PTMScan HS Ubiquitin/SUMO Remnant Motif Kit (Cat. No.: 59322; Cell Signaling Technologies) according to the manufacturer's instructions. The flow through (unbound peptides) from each sample was collected and used for whole-proteome profiling. Di-Gly enriched and total peptides were reconstituted in 200 mM EPPS, pH 8.5, for TMT labeling. Anhydrous acetonitrile was added to each sample to achieve a final concentration of 33% acetonitrile. Peptides from each sample were labeled with TMTpro reagents (Thermo Fisher Scientific) for 2 h at RT. Labeling reactions were quenched with 0.5% hydroxylamine and acidified with formic acid. Acidified peptides were combined and desalted by Sep-Pak (Waters).

TMT-labeled peptides were solubilized in 5% acetonitrile (ACN)/10 mM ammonium bicarbonate, pH 8.0, and ~300 *μg* of TMT-labeled peptides was separated by an Agilent 300 Extend C18 column (3.5-mm particles, 4.6-mm ID, and 250 mm in length). An Agilent 1260

binary pump coupled with a photodiode array (PDA) detector (Thermo Fisher Scientific) was used to separate the peptides. A 45-min linear gradient from 10% to 40% acetonitrile in 10 mM ammonium bicarbonate, pH 8.0 (flow rate of 0.6 ml/min), separated the peptide mixtures into a total of 96 fractions (36 s). A total of 96 fractions were consolidated into 24 samples in a checkerboard fashion and vacuum-dried to completion. All 24 fractions were desalted via Stage Tips and redissolved in 5% formic acid/5% acetonitrile for LC-MS3 analysis.

TMT-labeled di-Gly enriched peptides were fractionated using High pH Reversed-Phase Peptide Fractionation Kit (Cat. No.: 84868; Pierce). The 12 resulting fractions were compressed into six fractions. Each fraction was vacuum-dried to completion, desalted via Stage Tips, and redissolved in 5% formic acid/5% acetonitrile for LC-MS/MS analysis.

Total proteome data were collected on an Orbitrap Fusion Lumos mass spectrometer with a FAIMS device (Thermo Fisher Scientific) coupled to a Proxeon EASY-nLC 1000 LC pump (Thermo Fisher Scientific). Fractionated peptides were separated using a 90-min gradient at 500 nl/min on a 35-cm column (i.d. 100 *μm*, Accucore, 2.6 *μm*, 150 Å) packed in-house. FAIMS compensation voltages were set to –40, –60, and –80 with a 1.25-s cycle time. MS1 data were collected in the Orbitrap (60,000 resolution; maximum injection time 50 ms; AGC 4 × 105). Charge states between 2 and 5 were required for MS2 analysis, and a 120-s dynamic exclusion window was used. Top 10 MS2 scans were performed in the ion trap with CID fragmentation (isolation window 0.6 Da; NCE 35%; maximum injection time 35 ms; AGC 1 × 104). An online real-time search algorithm (Orbiter) was used to trigger MS3 scans for quantification (Schweppe et al, 2020).

MS3 scans were collected in the Orbitrap using a resolution of 50,000, NCE of 55%, maximum injection time of 200 ms, and AGC of 3.0 × 105. The closeout was set at two peptides per protein per fraction (Schweppe et al, 2020).

Ubiquitin data were collected on an Orbitrap Eclipse mass spectrometer with a FAIMS device enabled (Thermo Fisher Scientific) coupled to a Proxeon EASY-nLC 1200 LC pump (Thermo Fisher Scientific). Di-Gly enriched peptides were separated using a 90-min gradient at 575 nl/min. FAIMS compensation voltages were set to –40, –60, and –80 V with a cycle time of 1 s for the first shot and –45 and –65 V with a cycle time of 1.5 s for the second shot (Schweppe et al, 2019). MS1 data were collected in the Orbitrap (60,000 resolution; maximum injection time 50 ms; AGC 4 × 105). Charge states between 2 and 5 were required for MS2 analysis, and a 90-s dynamic exclusion window was used. MS2 scans were performed in the Orbitrap with HCD fragmentation (isolation window 0.7 Da; 50,000 resolution; NCE 36%; maximum injection time 150 ms; AGC 1.25 × 105).

Raw files were converted to mzXML, and monoisotopic peaks were reassigned using Monocle (Rad et al, 2021). Searches were performed using the Comet search algorithm against a human database downloaded from UniProt in May 2021. For total proteome searches, we used a 50 ppm precursor ion tolerance, 1.0005 fragment ion tolerance, and 0.4 fragment bin offset for MS2 scans collected in the ion trap. For di-Gly searches, we used a 50 ppm precursor ion tolerance and 0.02 fragment ion tolerance for MS2 scans collected in the Orbitrap. TMTpro on lysine residues and

peptide N-termini (+304.2071 Da) and carbamidomethylation of cysteine residues (+57.0215 Da) were set as static modifications, whereas oxidation of methionine residues (+15.9949 Da) was set as a variable modification. For di-Gly peptide analysis, +114.0429 Da was set as a variable modification on lysine residues.

Each run was filtered separately to a 1% false discovery rate (FDR) on the peptide/spectrum match level. Then, proteins were filtered to the target 1% FDR level across the entire combined dataset. For reporter ion quantification, a 0.003-Da window around the theoretical m/z of each reporter ion was scanned, and the most intense m/z was used. Reporter ion intensities were adjusted to correct for isotopic impurities of the different TMTpro reagents according to the manufacturer's specifications. Peptides were filtered to include only those with a summed signal-to-noise (SN) ratio ≥ 60 across all TMT channels. The signal-to-noise (S/N) measurements of peptides assigned to each protein were summed (for a given protein). These values were normalized so that the sum of the signal for all proteins in each channel was equivalent, thereby accounting for equal protein loading. The resulting normalization factors were used to normalize the ubiquitin sites, again to account for equal protein loading.

To compare between groups, normalization factors within groups were averaged and divided by the mean value for the protein or ubiquitinated peptide in non-targeting samples. A standard two-tailed t test yielded P-values. The log2 fold change was used as the measure of abundance change relative to non-targeting samples. For estimates of whole-protein ubiquitination, the SN values for individual ubiquitinated peptides corresponding to the same protein were added together before comparison.

## Subcellular fractionation mass spectrometry

Parental U2OS cells were split and grown separately in light (standard DMEM + dialyzed FBS + L-arginine and L-lysine) or heavy (DMEM + dialyzed FBS + 13C6 15N2 lysine and 13C6 15N4 arginine) media for 2 wk. During this time, cells were expanded to ~50 × 10^6 cells. Cells were then split into 15-cm plates for the intended experiment—16 samples, where each sample represents three 15-cm plates. 8 samples were "experimental" in nature and were forward-transfected with siNT (light) or siHAPSTR1 (heavy) for 16 h before harvest. Eight were "technical" in nature and were unperturbed to account for the technical variability of the assay and for unintended consequences of the separate culture in different medium formulations.

For subcellular fractionation, eight cell pellets were obtained by combining three heavy and eight light plates (one light and one heavy sample each, thus yielding a total of four experimental and four technical pellets). Pellets were then used for subcellular fractionation as described above. Cytoplasmic and nuclear fraction protein concentrations were determined by the BCA assay. 100 µg of each sample at 1 mg/ml was then precipitated using a chloroform–methanol protocol. Specifically, 400 µl of methanol was added to each sample, which was then vortexed before the addition of 100 µl chloroform. Samples were vortexed, and 300 µl H2O was added, vortexed again, and then spun at 14,000g × 1 min. The aqueous top layer was removed, then 400 µl methanol was added, and samples were vortexed, then spun again at 14,000g × 2 min.

Methanol was removed from the pellet, then pellets were dried on the benchtop, and samples were then stored at −80.

Four experimental cytoplasmic fractions with a 1:1 mix of siNT light and siHAPSTR1 heavy, four experimental nuclear fractions with a 1:1 mix of siNT light and siHAPSTR1 heavy, four cytoplasmic fraction technical controls with a 1:1 mix of WT light and WT heavy, and four nuclear fraction technical controls with a 1:1 mix of WT light and WT heavy were denatured with 8 M urea (in a 100 mM ammonium bicarbonate vortex for 1 h at RT) and processed with ProteaseMAX according to the manufacturer's protocol. The samples were reduced with 5 mM Tris(2-carboxyethyl)phosphine (TCEP; vortexed for 1 h at RT), alkylated in the dark with 10 mM iodoacetamide (IAA; 20 min at RT), diluted with 100 mM ammonium bicarbonate, and quenched with 25 mM TCEP. Samples were sequentially digested with Lys-C (2 h at 37°C with shaking) and trypsin (overnight at 37°C with shaking), acidified with TFA to a final concentration of 0.1%, and spun down (15,000g for 15 min at RT) wherein the supernatant was moved to a new tube. The samples were desalted using Peptide Desalting Spin Columns (89852; Pierce) and dried down with vacuum centrifugation. The samples were resuspended in 100 mM Hepes, pH 8.5, Micro BCA (23235; Pierce) assay was performed to determine the peptide concentration, and 46 µg of each sample was used for isobaric labeling. 16plex-TMT labeling was performed on the samples according to the manufacturer's instructions (A44520; Thermo Fisher Scientific). After incubation for 2 h at RT, the reaction was quenched with 0.3% (vol/vol) hydroxylamine, isobaric-labeled samples were combined 1:1:1:1:1:1:1:1:1:1:1:1:1:1:1:1, and the pooled sample was dried down with vacuum centrifugation. The sample was resuspended in 0.1% TFA solution, desalted using Peptide Desalting Spin Columns (89852; Pierce), and dried down with vacuum centrifugation. The sample was resuspended in 0.1% TFA solution and fractionated into eight fractions using High pH Reversed-Phase Peptide Fractionation Kit (84868; Pierce) wherein fractions were step-eluted in 300 µl of buffer of increasing ACN concentrations with decreasing concentrations of 0.1% triethylamine per the manufacturer's instructions. The fractions were dried down with vacuum centrifugation.

Two micrograms of each fraction or sample was autosampler-loaded with an UltiMate 3000 HPLC pump onto a vented Acclaim PepMap 100, 75 µm × 2 cm, nanoViper trap column coupled to a nanoViper analytical column (Cat#: 164570, 3 µm, 100 Å, C18, 0.075 mm, 500 mm; Thermo Fisher Scientific) with a stainless steel emitter tip assembled on Nanospray Flex Ion Source with a spray voltage of 2,000 V. Orbitrap Fusion (Thermo Fisher Scientific) was used to acquire all the MS spectral data. Buffer A contained 94.785% H2O with 5% ACN and 0.125% FA, and buffer B contained 99.875% ACN with 0.125% FA. The chromatographic run was for 4 h in total with the following profile: 0–7% for 7, 10% for 6, 25% for 160, 33% for 40, 50% for 7, 95% for 5, and again 95% for 15 min, receptively.

We used a MultiNotch MS3-based TMT method to analyze all the TMT samples (Ting et al, 2011; McAlister et al, 2014). The scan sequence began with an MS1 spectrum (Orbitrap analysis, resolution 120,000, 400–1,400 Th, AGC target 2 × 10^5, maximum injection time 200 ms); MS2 analysis, "Top speed" (2 s), collision-induced dissociation (CID, quadrupole ion trap analysis, AGC 4 × 10^3, NCE 35, maximum injection time 150 ms); and MS3 analysis, top 10 precursors, fragmented by HCD before Orbitrap analysis (NCE 55, max

AGC 5 × 10⁴, maximum injection time 250 ms, isolation specificity 0.5 Th, resolution 60,000).

Protein identification/quantification analyses were performed with Integrated Proteomics Pipeline (IP2; Bruker, Madison, WI.) using ProLuCID (Eng et al, 1994; Xu et al, 2015), DTASelect2 (Tabb et al, 2002; Cociorva et al, 2007), and Census and Quantitative Analysis. Spectrum raw files were extracted into MS1, MS2, and MS3 files using RawConverter (http://fields.scripps.edu/downloads.php). Independent searches were performed for light and heavy stable isotope labeling by amino acid in cell culture (Savitski et al, 2018). For the light search, cysteine carbamidomethylation (57.02146 C) and 16plex-TMT modification on lysine (304.2071 K) were set as static modifications. Acetylation (42.0106) and 16plex-TMT modification (304.2071) of N-termini were set as differential modifications. For the heavy search, cysteine carbamidomethylation, lysine 13C6 15N2 with 16plex-TMT (312.2213 K), and arginine 13C6 15N4 (10.0083 R) were set as fixed. Acetylation (42.0106) and 16plex-TMT modification (304.2071) of N-termini were set as differential modifications. The tandem mass spectra were searched against the UniProt human protein database (downloaded on 01-01-2014) (UniProt Consortium, 2015) and matched to sequences using the ProLuCID/SEQUEST algorithm (ProLuCID version 3.1) with 5 ppm peptide mass tolerance for precursor ions and 600 ppm for fragment ions. The search space included all fully and half-tryptic peptide candidates within the mass tolerance window with no-miscleavage constraint, which are assembled and filtered with DTASelect2 through IP2. To estimate peptide probabilities and FDR accurately, we used a target/decoy database containing the reversed sequences of all the proteins appended to the target database (Peng et al, 2003). Each protein identified was required to have a minimum of one peptide of minimal length of six amino acid residues; however, this peptide had to be an excellent match with an FDR < 1% and at least one excellent peptide match. After the peptide/spectrum matches were filtered, we estimated that the peptide FDRs were ≤ 1% for each sample analysis. Then, we used Census and Quantitative Analysis in IP2 for peptide quantification and normalization. Then, DTASelect2 was used to separate the heavy and light peptides. From the peptide data, we then quantified the amounts of heavy and light proteins. To avoid the recalculation of the same peptides mapped to different proteoforms, all peptides encoded by the same gene were grouped and quantified together. Spyder (MIT, Python 3.7, libraries, "numPy," "math") was used for the data analyses. The mass spectrometry data for this experiment was deposited in Mass Spectrometry Interactive Virtual Environment (MassIVE) under the identifier MSV000094168 and ProteomeXchange under the identifier PXD050150.

## Cell growth and fitness assays

Cell growth and migration assays in HeLa, 293T, and 231 cells were quantified using an IncuCyte live-cell imaging apparatus (Sartorius). For growth assays, the confluence percentage was tracked over time using optimized confluence masks for each cell line. For migration assays, 231 cells were plated at high density in 96-well plates and allowed to reach 100% confluence before wounding using the IncuCyte WoundMaker tool. Cells were then washed once with serum-free media and then plated in serum-free media to minimize proliferation over the assay time course. Migration was then quantified as the confluence of the wounded area over time.

Cell growth assays in MEFs used a Vi-CELL BLU cell counter (Beckman Coulter). 100,000 cells per sample were plated in a 12-well plate and then allowed to grow for either 2 or 3 d (based on the confluence of WT wells). Cells from all groups were then trypsinized, neutralized in DMEM, counted, and then replated at 100,000 cells per well. Cell counts over time were then compounded and visualized as population doublings.

## Cycloheximide (CHX) chase assays

Protein stability was assessed using CHX chase assays. Specifically, genetically manipulated 293T cells were plated at 800,000 cells per well in six-well plates the night before the intended assay. CHX was then diluted in DMEM and added to the plate to achieve a final concentration of 40 µg/ml for HAPSTR1 or 100 µg/ml for DDIT4 (higher concentration for a very short half-life protein). At the indicated timepoint, plates were washed immediately with cold PBS and then either frozen or lysed immediately in cold lysis buffer as described above.

## Immunofluorescence and aggregate analysis

Immunofluorescence was performed as previously described (Amici et al, 2022). Briefly, U2OS cells were reverse-transfected with the indicated siRNA, incubated for 48 h, and then plated onto poly-D-lysine–treated sterile coverslips in a 24-well plate. Fixation used 4% PFA. Anti-ubiquitin (FK2) primary and FITC-conjugated anti-mouse secondary antibodies were diluted 1:500 and 1:1,000, respectively, in 2% FBS. 10 blinded images were taken from three slides per condition. Aggregates were counted using an automated approach in Fiji. Specifically, DAPI images were converted to eight bits and thresholded to identify individual nuclei. Then, FITC signal was converted to eight bits, processed to find maxima that corresponded to aggregate foci (noise tolerance: 40), and counted per each nucleus.

## RNA-sequencing

RNA was extracted using the QIAGEN RNeasy kit including the optional on-column DNase treatment. Libraries were prepped using QuantSeq 3′ mRNA-Seq Library Prep Kit FWD for Illumina (Lexogen) as previously described (Smith et al, 2022). Libraries were then analyzed for quality using the Agilent High Sensitivity DNA kit and for quantity using Qubit dsDNA HS Assay. Libraries were then pooled and sequenced using NovaSeq 6000 SP Reagent Kit (100 cycles). Libraries were pooled and diluted to 1.25 nM, denatured with 1 M NaOH added to a 0.2 M final concentration (8 min at RT), and quenched with 400 mM Tris–HCl (pH 8). 5% PhiX spike-in (Illumina) was included. Pooled, denatured libraries were run on an Illumina NovaSeq using 76-bp reads, 12-bp index reads, and single-end single-read parameters. Bcl files were converted to FASTQ using bcl2fastq. Sequence quality was confirmed using FastQC v0.11.2. Trimming was performed using BBduk (BBTools v25.92). Alignment used STAR v2.6.0 and reference genomes hg38 and mm9. Differential expression analyses used EdgeR v3.26.8. The

analysis in Fig 3E used data originally published previously (Amici et al, 2022), which were reanalyzed as discussed in the main text. Differentially expressed genes were only considered with adjusted *P*-values < 0.05.

## Gene essentiality data

Gene essentiality data were obtained from the Dependency Map (DepMap) portal (https://depmap.org/portal/download/—21q4 release). Chronos scores were used to quantify the fitness effect of individual gene loss, with "essentiality scores" in this study represented as the Chronos score multiplied by –1. For example, a highly essential gene might have a Chronos fitness effect of –1.0 and thus an essentiality score of 1.0. For analyses of HAPSTR1 essentiality in p53-intact versus p53-lost background, we used p53 essentiality scores instead of available hotspot mutation data because of the myriad mechanisms by which cells inactivate p53. That is, we grouped cells that grow markedly faster after p53 KO together as p53-intact and cells that did not have any growth effect from p53 KO together as p53-lost. Coessentiality analysis was performed using FIREWORKS (https://fireworks.mendillolab.org) as previously described (Amici et al, 2021).

## Gene set enrichment analysis

Gene set enrichment analysis (Subramanian et al, 2005) was performed using the Molecular Signature DataBase (Liberzon et al, 2011) as accessible at software.broadinstitute.org/gsea/msigdb/annotate.jsp. The gene sets queried were as follows: hallmark (H) for human or orthology-mapped hallmark for mouse (MH), chemical and genetic perturbations (CGP), REACTOME canonical pathways (CP:REACTOME), and gene ontology biological process (GO:BP), molecular function (GO:MF), and cellular compartment (GO:MF). Gene sets highlighted in the work all had enrichment FDR of less than $1 \times 10^{-5}$. Genes/proteins selected for each enrichment analysis comprised either all hits from the given analysis (if less than 200) or the top 200 hits, sorted by multiple-comparison adjusted *P*-value.

## Mouse generation and breeding strategies

All mouse experiments were performed in accordance with institutional guidelines under protocol ID IS00016896.

Generation of HAPSTR1-floxed mice was performed by the Ingenious targeting library. Briefly, a 12.4-kb region used to construct the targeting vector was subcloned from a positively identified C57BL/6 BAC clone (RP23-65I18) using homologous recombination-based techniques. The targeting vector was constructed such that a loxP-F3–flanked hygromycin cassette is placed 2,160 bp upstream of exon 1. A loxP-FRT–flanked Neo selection cassette is placed 1,079 bp downstream of exon 2. The middle arm that contains the region to be floxed and thus excised upon Cre recombination is 4.5 kb and contains exons 1 and 2. The 5' homology arm of the vector was ~6 kb in length, and the 3' homology arm was ~4.5 kb. The targeting vector was confirmed by restriction analysis and sequencing after each modification step. The vector was thoroughly validated by Sanger sequencing. The BAC was subcloned into an iTL cloning vector

(~3 kb), derived from a pBR322 (Promega) backbone vector containing an ampicillin selection cassette for retransformation of the construct before electroporation. A loxP-FRT-hUBS-gb2-Neomycin-FRT cassette was inserted. The targeting construct was then linearized using Srf I before electroporation into embryonic stem cells.

Founder mice were identified containing the desired loxP sites and then transferred to Northwestern University for quarantine and further breeding. Mice were inbred, then backcrossed with standard WT C57BL/6J mice (Jax strain 000664). The presence of the floxed allele and the absence of the FLP recombinase (present in one founder) by PCR were criteria to allow further breeding (see below). Hapstr1-floxed mice were then crossed with mice containing a Sox2Cre allele (Jax strain 008454) or a CreERT2 (Jax strain 004682) allele. Hapstr1-floxed Sox2Cre mice were further crossed with a germline p53 knockout strain (Jax strain 002101).

## Mouse genotyping

For PCR, the CloneAmp HiFi master mix was used along with primers at a final concentration of 0.3 $\mu$M. PCR primers for genotyping are listed in the Key Resources table. Genotyping was typically performed using an optimized touchdown PCR protocol: first, 94°C x 2 min; second, 10 cycles of (94°C × 10 s, 65°C* × 5 s, 68.5°C × 5 s) where *indicates a decrease of 0.5°C per cycle; and last, 26 cycles of (94°C × 10 s, 60°C × 5 s, 72°C × 5 s). Positive and negative controls for the desired genotypes were used to confirm PCR efficacy. Note that Hapstr1 has been genotyped using different primers over time because of further optimization of the process. Currently, we recommend primers 60 and 61 for genotyping the WT and floxed alleles (169-bp band for WT allele, 245-bp band for floxed allele), and 60 and 218 for the knockout allele (500-bp band if loxP-mediated knockout has occurred).

## Oligo sequence

58 FLP F ACAGAGACAAAGACAAGCGTTAGTAGG.
59 FLP R ATTTCCCACAACATTAGTCAACTCCGTTAGG.
60 H1 shared F CCCAAGGCAGAGAAACTCCT.
61 H1 WT R ACCAGGACTTCATAGGCAGA.
65 H1 KO R1 AGACTGGACTGCGTTTCTCA.
218 H1 KO R2 ATCCGATGCTGTCTTCTGGT.
Cre F GCTAACCATGTTCATGCCTTC.

## Observed versus expected knockout rates

For all breeding pairs where it was possible to observe a homozygous knockout among the offspring, we calculated the Mendelian probability of observing each genotype. For these calculations, it is important to note that female Sox2Cre+ mice will result in Cre activity in the early embryo regardless of whether the actual Cre allele is inherited, whereas offspring from a Sox2Cre+ male will only have Cre activity if they inherit the allele. Thus, from parent Hapstr1 and Cre genotypes, the probability of WT (any combination of floxed and WT alleles), heterozygote (ko/wt or ko/fl), or homozygous knockout can be calculated and multiplied by the litter size to yield the expected number of knockouts for that

litter. Observed genotypes were then tracked for each litter over time. A binomial test implemented in scipy.stats was then used to determine the *P*-value for the number of a given genotypes observed versus expected. Based on the lack of observed knockouts, the expected observation rate for heterozygotes was calculated under the assumption that knockouts were not a possible outcome for mice in a given litter. An analogous approach was used to determine expected knockouts in each timed breeding embryo harvest (expected Mendelian proportions x number of animals identified). For the p53 cross-experiment, only the mice in a given litter that were confirmed to be homozygous p53 KO were used in the analysis of Hapstr1 genotype frequencies.

### Induced knockout of Hapstr1 in adult mice

Litters containing wt/wt, fl/wt, and fl/fl genotypes for Hapstr1 (in the presence or absence of Cre) were used for induced knockout experiments. Mice were allowed to develop fully and reach the age of 3 mo before beginning the experiment. Tamoxifen was then administered via intraperitoneal (IP) injection, delivering 150 $\mu$l to each mouse for four straight days, after which standard chow was replaced with tamoxifen chow (Teklad TD.130855; 250 mg tamoxifen/kg). Weights and appearance were tracked over time for all mice.

### Chemical hair removal and quantification of regrowth

Nair was applied to the hindlimb using a Q-tip 2 wk before scheduled euthanasia for the induced knockout experiment. After euthanasia, limbs were removed and a photo was taken with all limbs adjacent on a white piece of paper. The image was then converted to grayscale, and the mean gray value (0–255 scale) for the treated region was quantified (ImageJ). Values were then subtracted from 255 (scale max) to yield a value corresponding to how dark the area was (indicating hair regrowth). Knockout scores were represented as a proportion of the average WT mouse limb.

### Mouse organ harvest, fixation, and histology

Mice were euthanized by primary cervical dislocation followed immediately by decapitation. Organs were then surgically isolated, washed in PBS, and either homogenized in lysis buffer (described above) using Tissue-Tearor for protein analysis or submerged in 10% formalin (for histology). Tissues were formalin-fixed for 72 h on a shaker at low speed at RT before transfer to 70% ethanol and submission to the Mouse Histology & Phenotyping Laboratory for paraffin embedding, sectioning, and H&E staining. Slides were reviewed by an independent pathologist.

## Supplementary Information

## Acknowledgements

Total and diglycine proteomic experiments were performed in collaboration with the Thermo Fisher Scientific Center for Multiplexed Proteomics at Harvard Medical School (https://tcmp.hms.harvard.edu). Histology services were provided by the Northwestern University Mouse Histology and Phenotyping Laboratory, which is supported by NCI P30-CA060553 awarded to the Robert H Lurie Comprehensive Cancer Center. This work was supported by NIHF30CA264513 and NIHT32GM008152 to DR Amici, NIHR01AG078796 to JN Savas, and NIH1R01GM144617-01 to ML Mendillo, and American Cancer Society ABOA Impact RSG-22-086-01-TBE to ML Mendillo.

## Author Contributions

DR Amici: conceptualization, resources, data curation, software, formal analysis, funding acquisition, investigation, visualization, methodology, and writing—original draft, review, and editing.
S Alhayek: data curation, formal analysis, validation, visualization, methodology, and writing—review and editing.
AT Klein: data curation, formal analysis, and writing—review and editing.
Y-Z Wang: data curation, formal analysis, and writing—review and editing.
AP Wilen: investigation, methodology, and writing—review and editing.
W Song: investigation, methodology, and writing—review and editing.
P Zhu: investigation, methodology, and writing—review and editing.
A Thakkar: investigation, methodology, and writing—review and editing.
MA King: investigation and writing—review and editing.
AWT Steffeck: methodology and writing—review and editing.
MJ Alasady: software and writing—review and editing.
C Peek: methodology and writing—review and editing.
JN Savas: resources, supervision, funding acquisition, methodology, project administration, and writing—review and editing.
ML Mendillo: conceptualization, resources, supervision, funding acquisition, methodology, project administration, and writing—review and editing.

## Conflict of Interest Statement

The authors declare that they have no conflict of interest.

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
