## [Reviewer comments · Life Science Alliance]

Life Science Alliance

Tight regulation of a nuclear HAPSTR1-HUWE1 pathway essential for mammalian life

David Amici, Sammy Alhayek, Austin Klein, Yi-Zhi Wang, Anika Wilen, Weimin Song, Pei Zhu, Abhishek Thakkar, McKenzi King, Adam Steffek, Milad Alasady, Clara Peek, Jeffrey Savas, and Marc Mendillo

DOI: <https://doi.org/10.26508/lsa.202302370>

Corresponding author(s): Marc Mendillo, Northwestern University and Marc Mendillo, Northwestern University

Review Timeline:

Submission Date:	2023-09-13
Editorial Decision:	2023-10-23
Revision Received:	2024-01-26
Editorial Decision:	2024-02-15
Revision Received:	2024-02-26
Accepted:	2024-02-27

Transaction Report:

October 23, 2023

Re: Life Science Alliance manuscript #LSA-2023-02370-T

Marc L Mendillo
Northwestern University Feinberg School of Medicine
303 E. Superior St., Room 7-303
Chicago, Illinois 60610

Dear Dr. Mendillo,

Thank you for submitting your manuscript entitled "Tight regulation of a nuclear HAPSTR1-HUWE1 pathway essential for mammalian life" to Life Science Alliance. The manuscript was assessed by expert reviewers, whose comments are appended to this letter. We invite you to submit a revised manuscript addressing the Reviewer comments.

Thank you for this interesting contribution to Life Science Alliance. We are looking forward to receiving your revised manuscript.

Sincerely,

B. MANUSCRIPT ORGANIZATION AND FORMATTING:

Reviewer #1 (Comments to the Authors (Required)):

The authors have previously identified HAPSTR1 as a molecular rheostat within an integrated network of stress response pathways. Its effective modulation of stress signaling requires a ubiquitin E3 ligase, HUWE1 (PMID: 35776542). In this study, the authors further investigated the biochemical characteristics and physiological functions of the HAPSTR1-HUWE1 pathway. They found that the maintenance of HAPSTR1 homeostasis is critical for cell fitness, likely by modulating the localization of HUWE1. Moreover, the authors identified ubiquitin E3 ligase TRIP12 and deubiquitinase USP7 as upstream regulators of HAPSTR1's stability. Finally, they generated HAPSTR1 conditional KO mice to address the physiological functions of HAPSTR1 in vivo. Overall, this study provides insights into the molecular regulation and functions of HAPSTR1, particularly its role in regulating nuclear HUWE1. Considering the recent report (PMID: 37167062) demonstrating that HAPSTR1 facilitates the nuclear localization of HUWE1, the novelty presented in this work is somewhat compromised. Moreover, the exact mechanism by which HAPSTR1 regulates HUWE1 activity/function and how HAPSTR1-HUWE1 pathway exerts its effects on cell growth via p53/p21 remain unclear in this work. Direct experimental evidence supporting their hypothesis is recommended before considering it for publication.

Major points

1. How does HAPSTR1 regulate HUWE1 ubiquitin E3 ligase activity and its HAPSTR1-dependent function?
 - a. HUWE1-dependent ubiquitination does not require HAPSTR1 (Fig. 1f), whereas suppression of either HAPSTR1 or HUWE1 results in comparable consequences in proteomic changes (Fig. 1c). How do the authors interpret these results? How does HAPSTR1 regulate HUWE1 ubiquitin E3 ligase activity?
 - b. If the hypothesis of "too many inhibiting aspects of HUWE1 function, and too little preventing HUWE1 function in the nucleus" holds true, are the affected targets identified after either HAPSTR1 or HUWE1 knockdown in Fig. 1c mostly nuclear proteins?
 - c. In Fig. 2d, HAPSTR1 is shown to be in the cytosol in unperturbed conditions. If it is a cytosolic protein, how can it directly regulate nuclear localization of HUWE1?
 - d. How does HAPSTR1 overexpression affect HUWE1 ubiquitin E3 ligase activity? Fig. 3g shows that HAPSTR1 overexpression does not affect degradation of DDIT4, a known substrate of HUWE1. However, HAPSTR1 overexpression leads to a much higher level of nuclear HUWE1 while the cytosolic HUWE1 level is similar to that in control (Fig. 2d). Does that mean that HAPSTR1 overexpression-mediated increase in nuclear HUWE1 does not possess its canonical ubiquitin E3 ligase activity?
 - e. Is the reduce of cell fitness after HAPSTR1 overexpression dependent on HUWE1? Is it through modulation of HUWE1 ubiquitin E3 ligase activity/localization or unidentified mechanisms independent of HUWE1?
 - f. The current work proposes that HAPSTR1 mediates the nuclear localization of HUWE1, aligning with a recent work by Monda et al (PMID: 37167062). However, there is a discrepancy regarding the functions of nuclear HAPSTR1-HUWE1 in these two works. Monda et al showed that HAPSTR1 is required for degradation of a subset of nuclear-localized HUWE1 substrates, whereas the current work showed that the function of HAPSTR1-HUWE1 is not mediated through HUWE1's canonical ubiquitin E3 ligase activity (Fig. 3g). How can this discrepancy be explained?
2. How is the HAPSTR1-HUWE1 pathway linked to the p53 pathway?
 - a. Is the slow growth in HAPSTR1 KO MEF dependent on p53? Is the elevated p53 in HAPSTR1 KO MEF (Fig. 5h) a result of decreased HUWE1 activity or decreased nuclear HUWE1 level?
 - b. Is the effect of HAPSTR1 overexpression on cell fitness (fig. 3) also dependent on p53?
 - c. Previously, HUWE1 has been shown to ubiquitinate p53 in vitro (PMID: 15989956). What would be the potential role of HAPSTR1 in mediating HUWE1-dependent ubiquitination of p53 (i.e., catalyze p53 ubiquitination, or nuclear localization of HUWE1)?
3. To what extent can the observed phenotypes in HAPSTR1 KO MEF be attributed to HAPSTR1-HUWE1 and HAPSTR1-HUWE1-p53 pathway? Authors may consider knockdown HUWE1 and p53 in HAPSTR1 KO MEF and compare their respective doublings.

Minor points

1. How did the authors carry out the analysis of DNA damage screen for HAPSTR1 coessentiality in fig. 4a, right panel?
2. What could be the mechanism underlying the HAPSTR1 KO-mediated decrease in cell fitness? In fig 5f, growth is similar to WT for the first few passages, then slows down from passage 5 onwards. Is this suppression of cell growth in HAPSTR1 KO MEF dependent on p53? If yes, it is also puzzling since p53 is shown to be elevated in HAPSTR1 knockout cells as early as passage 2 (fig. 5h).
3. Typo: ... the vast majority did not appear to be ubiquitinated by or otherwise functionally related to HUWE1 in our datasets.

Finally, we investigated whether we could identify global alterations in "polyubiquitin" linkages

Reviewer #2 (Comments to the Authors (Required)):

In this manuscript, the authors define a functional relationship between HAPSTR1 and the E3 ligase HUWE1. They show that HAPSTR1 contributes to promoting nuclear localization of HUWE1 to effect some aspects of cellular signaling. Both overexpression or depletion of HAPSTR1 modestly impacted cellular fitness and that loss of HUWE1 showed similar transcriptional changes to that observed upon HAPSTR1 overexpression. This appears to result from direct interactions between HUWE1 and HAPSTR1 as mutations in HAPSTR1 that disrupt interactions between these proteins do not lead to changes in transcript levels. They go on to identify the Ub ligase TRIP12 and the deubiquitinase USP7 as potential regulators of HAPSTR1 stability. The authors then create a conditional HAPSTR1-deficient mouse. Deletion of HAPSTR1 early in embryogenesis did not yield viable animals owing to various proposed factors, largely attributed to reduced HUWE1 nuclear activity. They then went on to create partial inducible HAPSTR1 knockouts in 3 month animals and showed that these mice demonstrated some phenotypes, including reduced weight (potentially due to digestive distress) and reduced hair growth (reflecting impaired reparative processes).

This manuscript is focused on defining the molecular relationship between HAPSTR1 and HUWE1. While some of the experiments are well performed, numerous experimental efforts are 'cut short', limiting the potential impact of the work. For example, while the authors have somewhat strong data showing the importance of HAPSTR1 for nuclear HUWE1 localization (a result that unfortunately was also recently published by another group during this work), there is little description of how this activity impacts HUWE1 activity. They suggest that this effect appears to be largely independent of HUWE1 ubiquitin ligase activity, but what is HUWE1 doing then? There are some experiments looking into the regulation of HAPSTR1 by TRIP12 and USP7, but, while encouraging, the molecular basis of this effect is not well demonstrated (needs rep/quant/stats) and not well explained. Finally, they make a conditional knockout mouse (which I'm certain will be well received by the community), but don't really take it that much farther. Specific questions include would heterozygous Hapstr1 deletion mice appear to show similar reductions in HAPSTR1 to the iKO mice used in Figure 6 show similar phenotypes? This is ultimately a difficult manuscript to review, as there are numerous interesting discoveries made. I would like to see some type of resolution for one of the three above questions prior to publication. Some additional depth to accompany the breadth of this study. There are many suggestions to explain these various results discussed in the manuscript, but some experiments to probe these models deeper would significantly improve this current manuscript. That being said, improving some of the experiments describing the regulation of HAPSTR1 by USP7 and TRIP12 (reps/quant/stats) could strengthen this paper sufficiently to publish in LSA as 'descriptive data'.

Additional Points.

Figure 2d. No endogenous HAPSTR1 in nuclear fraction, is that because of detection? Also, nuclear localization of HUWE1 is not the most convincing in this figure.

Figure 4d needs quantification across different conditions to convince more efficiently.

Page 11 second paragraph. I think the Figure 3D callout should be Figure 4D.

Reviewer #1 (Comments to the Authors (Required)):

The authors have previously identified HAPSTR1 as a molecular rheostat within an integrated network of stress response pathways. Its effective modulation of stress signaling requires a ubiquitin E3 ligase, HUWE1 (PMID: 35776542). In this study, the authors further investigated the biochemical characteristics and physiological functions of the HAPSTR1-HUWE1 pathway. They found that the maintenance of HAPSTR1 homeostasis is critical for cell fitness, likely by modulating the localization of HUWE1. Moreover, the authors identified ubiquitin E3 ligase TRIP12 and deubiquitinase USP7 as upstream regulators of HAPSTR1's stability. Finally, they generated HAPSTR1 conditional KO mice to address the physiological functions of HAPSTR1 in vivo. Overall, this study provides insights into the molecular regulation and functions of HAPSTR1, particularly its role in regulating nuclear HUWE1. Considering the recent report (PMID: 37167062) demonstrating that HAPSTR1 facilitates the nuclear localization of HUWE1, the novelty presented in this work is somewhat compromised. Moreover, the exact mechanism by which HAPSTR1 regulates HUWE1 activity/function and how HAPSTR1-HUWE1 pathway exerts its effects on cell growth via p53/p21 remain unclear in this work. Direct experimental evidence supporting their hypothesis is recommended before considering it for publication.

We appreciate the reviewer's assessment of the paper and have added substantial additional data to further improve our manuscript.

Major points

1. How does HAPSTR1 regulate HUWE1 ubiquitin E3 ligase activity and its HAPSTR1-dependent function?
 - a. HUWE1-dependent ubiquitination does not require HAPSTR1 (Fig. 1f), whereas suppression of either HAPSTR1 or HUWE1 results in comparable consequences in proteomic changes (Fig. 1c). How do the authors interpret these results? How does HAPSTR1 regulate HUWE1 ubiquitin E3 ligase activity?

We interpret the discordance between ubiquitin proteomic and other less targeted approaches (total proteomics, cell cycle measurements, RNA-seq) as reflective of a more specific assay for E3 ligase activity vs. less specific readouts of cell state. For example, HAPSTR1 or HUWE1 loss both activate p53 and suppress cell cycle progression, leading to differential expression of cell cycle phase-related proteins in total proteomics and genes in RNA-seq. Specific to the ubiquitin data, we interpret HUWE1-HAPSTR1 discordance as reflecting the difference between canonical, HAPSTR1-independent HUWE1 E3 ligase activity (recognizing certain primarily cytoplasmic target proteins) vs the postulated non-canonical role for HUWE1 in the nucleus. One explanation for this non-canonical activity is the suggestion from multiple recent reports that HUWE1 recognizes ubiquitinated proteins non-specifically to facilitate their polyubiquitination and clearance (in particular, see recent preprint doi.org/10.1101/2023.05.30.542866, as well as yeast work PMID: 34764209). Such an effect would not be well-detected by diglycine proteomics, which looks at the initial ubiquitin linkage rather than length or diversity of chain. Supporting a model where HAPSTR1 enables ubiquitin-directed ligase activity in the nucleus, we have added data which show that loss of either HUWE1 or HAPSTR1 increases the number of ubiquitinated protein aggregates in the nucleus upon heat stress (Fig. 2e, S2c). We have also edited the text to improve clarity of our suggested model.

b. If the hypothesis of "too many inhibiting aspects of HUWE1 function, and too little preventing HUWE1 function in the nucleus" holds true, are the affected targets identified after either HAPSTR1 or HUWE1 knockdown in Fig. 1c mostly nuclear proteins?

The proteins increased after either HUWE1 or HAPSTR1 loss indeed are strongly enriched for nuclear localization. We are careful not to imply these proteins are all direct substrates of nuclear HUWE1, but they likely include substrates as well as proteins increased indirectly as a result of nuclear HUWE1 loss.

c. In Fig. 2d, HAPSTR1 is shown to be in the cytosol in unperturbed conditions. If it is a cytosolic protein, how can it directly regulate nuclear localization of HUWE1?

Prior immunofluorescence, immunoprecipitation and proximity ligation proteomic evidence support HAPSTR1 as a protein which interacts with components of the nuclear pore and shuttles between cytosol and nucleus. Additionally, HAPSTR1 has a structural motif thought to contribute to membrane binding (amphipathic helix) and which is required for HUWE1-binding (Amici 2022). As such, we postulate that HAPSTR1 shuttles rapidly from cytosol to nucleus and that this membrane-binding activity is involved in modulation of HUWE1 localization. However, future structural, mutagenesis, and reconstitution studies will be necessary to fully understand this activity.

d. How does HAPSTR1 overexpression affect HUWE1 ubiquitin E3 ligase activity? Fig. 3g shows that HAPSTR1 overexpression does not affect degradation of DDIT4, a known substrate of HUWE1. However, HAPSTR1 overexpression leads to a much higher level of nuclear HUWE1 while the cytosolic HUWE1 level is similar to that in control (Fig. 2d). Does that mean that HAPSTR1 overexpression-mediated increase in nuclear HUWE1 does not possess its canonical ubiquitin E3 ligase activity?

Unfortunately, there are no suitable model nuclear substrates to use in assays for HUWE1's canonical E3 ligase activity (ie, well-defined nuclear proteins which HUWE1 reproducibly modifies). We and others have struggled reproducing HUWE1 effects on many reported nuclear substrate proteins in our systems. This makes answering the question of canonical HUWE1 activity in the nucleus somewhat difficult. While possible that HUWE1 does have canonical (ie, modification-independent, substrate-directed) E3 activity in the nucleus, our data suggest that the dominant effect is more likely non-canonical effects on protein quality control and cell cycle progression.

e. Is the reduce of cell fitness after HAPSTR1 overexpression dependent on HUWE1? Is it through modulation of HUWE1 ubiquitin E3 ligase activity/localization or unidentified mechanisms independent of HUWE1?

HUWE1 is among the most essential genes in the genome. As such, we have been unable to successfully knock out HUWE1 for use in epistatic growth experiments. Additionally, available HUWE1 "inhibitors" are nonspecific and toxic. Therefore, the best we can do is unbiased, high-sensitivity signaling experiments (i.e., RNAseq). These show HAPSTR1 overexpression has substantial effects on signaling, but that that HAPSTR1 overexpression has no effect if HUWE1 is not present. This allows us to confidently state that HUWE1 mediates much (if not all) of HAPSTR1 overexpression's phenotypic effects, at least in

the models that we have investigated thus far.

f. The current work proposes that HAPSTR1 mediates the nuclear localization of HUWE1, aligning with a recent work by Monda et al (PMID: 37167062). However, there is a discrepancy regarding the functions of nuclear HAPSTR1-HUWE1 in these two works. Monda et al showed that HAPSTR1 is required for degradation of a subset of nuclear-localized HUWE1 substrates, whereas the current work showed that the function of HAPSTR1-HUWE1 is not mediated through HUWE1's canonical ubiquitin E3 ligase activity (Fig. 3g). How can this discrepancy be explained?

We thoroughly enjoyed and appreciate the work of Monda et al. One aspect of their data which we interpret differently is in the conclusion that HAPSTR1 mediates degradation of specific nuclear substrates. Monda et al demonstrate that certain nuclear proteins are regulated by both HAPSTR1 and HUWE1. However, they do not provide evidence that HUWE1 ubiquitinates these proteins. We agree that many proteins are dual regulated by HAPSTR1 and HUWE1. However, given our data, we postulate that these might be indirect effects (i.e., that these proteins are increased by loss of the shared HAPSTR1-HUWE1 pathway function. We have also not observed effects on these specific proteins in our dataset (though we acknowledge using a different model system).

2. How is the HAPSTR1-HUWE1 pathway linked to the p53 pathway?

a. Is the slow growth in HAPSTR1 KO MEF dependent on p53? Is the elevated p53 in HAPSTR1 KO MEF (Fig. 5h) a result of decreased HUWE1 activity or decreased nuclear HUWE1 level?

3. To what extent can the observed phenotypes in HAPSTR1 KO MEF be attributed to HAPSTR1-HUWE1 and HAPSTR1-HUWE1-p53 pathway? Authors may consider knockdown HUWE1 and p53 in HAPSTR1 KO MEF and compare their respective doublings.

We have bundled question 2 with the major point in question 3 as they are related. To address this question, we have performed the suggested experiment of knocking out p53 in WT vs. HAPSTR1-KO MEFs. We observed that p53 knockout markedly increased the growth of both WT and HAPSTR1-KO MEFs and provided a partial (but incomplete) phenotypic rescue. This is consistent with additional data we have added demonstrating that HAPSTR1 is less essential, but still important for cell growth in cancer cell lines which lack p53 signaling. Finally, we are happy to finally be able to include an experiment long in the making. We crossed our germline knockout HAPSTR1 mice with p53-null mice, finding that p53 loss does not rescue the *in vivo* lethality phenotype. Altogether, these data answer a major question in the emerging HAPSTR1 field – whether p53 is the mediator of HAPSTR1/HUWE1 loss phenotypes.

b. Is the effect of HAPSTR1 overexpression on cell fitness (fig. 3) also dependent on p53?

This is not dependent on p53 as the cell lines used for these assays do not have intact p53 signaling. This is consistent with the additional experiments above which suggest that HAPSTR1 effects on p53 are indirect. We have clarified this in the text.

c. Previously, HUWE1 has been shown to ubiquitinate p53 *in vitro* (PMID: 15989956). What would be the potential role of HAPSTR1 in mediating HUWE1-dependent ubiquitination of p53 (i.e., catalyze p53 ubiquitination, or nuclear localization of HUWE1)?

Loss of HUWE1 and HAPSTR1 clearly results in increased p53 levels in p53-WT cells, and overexpression of HAPSTR1 suppresses p53 in a manner requiring HAPSTR1's nuclear localization ability (Amici 2022, Monda 2023, Lu 2022 BioRxiv). However, we are uncertain that this is related to a direct ubiquitination activity of HUWE1. We (and multiple other groups, via personal communications) have not found a robust and generalizable effect of HUWE1 on p53 ubiquitination in physiological settings. Regarding the referenced paper, it is increasingly appreciated that incubation of the HECT domain of HUWE1 alone (much easier to purify than the intact 500 kDa protein) can lead to non-specific *in vitro* ubiquitination activity for proteins which are not robust HUWE1 substrates in living cells (PMID: 34314700). This technical issue has been a significant problem in the field, where many reported substrates (validated by *in vitro* ubiquitination using HECT-only HUWE1) are not robustly or consistently modified by HUWE1 *in vivo* or using full length HUWE1 protein.

Minor points

1. How did the authors carry out the analysis of DNA damage screen for HAPSTR1 coessentiality in fig. 4a, right panel?

We applied the same approach as used with the DepMap data – pearson correlation based on gene-level fitness scores. In the case of the DNA damage screens, fitness scores were DrugZ scores (dropout relative to no drug) rather than Chronos scores used in DepMap (dropout in general). We have expanded the methods section about this.

2. What could be the mechanism underlying the HAPSTR1 KO-mediated decrease in cell fitness? In fig 5f, growth is similar to WT for the first few passages, then slows down from passage 5 onwards. Is this suppression of cell growth in HAPSTR1 KO MEF dependent on p53? If yes, it is also puzzling since p53 is shown to be elevated in HAPSTR1 knockout cells as early as passage 2 (fig. 5h).

The p53 epistasis data discussed above suggest that p53 mediates some of the growth effects attributable to HAPSTR1 loss, but does not entirely explain the growth deficits or perinatal lethality associated with HAPSTR1 loss. The RNAseq provides several additional candidates for altered signaling pathways (eg, TGFB, oxidative stress), but no candidate specific target. Given the profound signaling changes even with acute protein depletion, we believe it is likely that the loss of nuclear HUWE1 has pleiotropic effects (including activation of p53) by impacting HUWE1's ability to ubiquitinate proteins in the nucleoplasm and affect chromatin/DNA repair dynamics.

3. Typo: ... the vast majority did not appear to be ubiquitinated by or otherwise functionally related to HUWE1 in our datasets. Finally, we investigated whether we could identify global alterations in "polyubiquitthulin" linkages

Thank you we have fixed this typo.

Reviewer #2 (Comments to the Authors (Required)):

In this manuscript, the authors define a functional relationship between HAPSTR1 and the E3 ligase HUWE1. They show that HAPSTR1 contributes to promoting nuclear localization of HUWE1 to effect some aspects of cellular signaling. Both overexpression or depletion of HAPSTR1 modestly impacted cellular fitness and that loss of HUWE1 showed similar transcriptional changes to that observed upon HAPSTR1 overexpression. This appears to result from direct interactions between HUWE1 and HAPSTR1 as mutations in HAPSTR1 that disrupt interactions between these proteins do not lead to changes in transcript levels. They go on to identify the Ub ligase TRIP12 and the deubiquitinase USP7 as potential regulators of HAPSTR1 stability. The authors then create a conditional HAPSTR1-deficient mouse. Deletion of HAPSTR1 early in embryogenesis did not yield viable animals owing to various proposed factors, largely attributed to reduced HUWE1 nuclear activity. They then went on to create partial inducible HAPSTR1 knockouts in 3 month animals and showed that these mice demonstrated some phenotypes, including reduced weight (potentially due to digestive distress) and reduced hair growth (reflecting impaired reparative processes).

We appreciate the efforts of the reviewer in reviewing this paper.

This manuscript is focused on defining the molecular relationship between HAPSTR1 and HUWE1. While some of the experiments are well performed, numerous experimental efforts are 'cut short', limiting the potential impact of the work. For example, while the authors have somewhat strong data showing the importance of HAPSTR1 for nuclear HUWE1 localization (a result that unfortunately was also recently published by another group during this work), there is little description of how this activity impacts HUWE1 activity. They suggest that this effect appears to be largely independent of HUWE1 ubiquitin ligase activity, but what is HUWE1 doing then?

We appreciate the lack of a satisfying direct activity of nuclear HUWE1. This was not for a lack of effort, as we spent multiple years trying to identify robust nuclear HUWE1/HAPSTR1 substrates! At some point, we have begun to believe that the absence of evidence for specific shared substrates points toward evidence of absence. That is, HUWE1 appears to have very few robust substrates which it modifies reproducibly in multiple cell types – the only ones we have found are HAPSTR1 and DDIT4. On the other hand, emerging evidence suggests that HUWE1 has a great deal of importance via a less-specific ubiquitin ligase activity. For example, HUWE1 recognizes ubiquitinated proteins non-specifically to facilitate their polyubiquitination and clearance ([doi.org/10.1101/2023.05.30.542866](https://doi.org/10.1101/2023.05.30.542866)), as does ancestral Tom1 (yeast HUWE1; PMID: 34764209). In line with this model, we have added data in the updated manuscript showing that loss of either HAPSTR1 or HUWE1 results in an increased number of ubiquitinated nuclear aggregates after heat stress. We feel that this, in combination with our proteomic data, fits best a model where HUWE1 accesses the nucleus in a HAPSTR1-dependent manner and can then perform quality control functions on partially ubiquitinated or damaged proteins and augment cell cycle progression.

There is some experiments looking into the regulation of HAPSTR1 by TRIP12 and USP7, but, while encouraging, the molecular basis of this effect is not well demonstrated (needs rep/quant/stats) and not well explained.

We have added additional replicates and quantification for these experiments, established reproducible half-lives for each modification, and expanded the explanations of the experiments.

Finally, they make a conditional knockout mouse (which I'm certain will be well received by the community), but don't really take it that much farther. Specific questions include would heterozygous Hapstr1 deletion mice appear to show similar reductions in HAPSTR1 to the iKO mice used in Figure 6 show similar phenotypes?

We appreciate the acknowledgement that the mice created will be of great use to the community. To the point of not taking the mouse analyses farther, we have added additional data characterizing these mice. For example, we more concretely identify the point in development where HAPSTR1-null mice falter (~E15-16). Additionally, we have performed a significant *in vivo* epistasis experiment with these mice to answer a major question in the emerging HAPSTR1 field – to what extent p53 mediates HAPSTR1 phenotypes. We crossed HAPSTR1-null and p53-null mice, demonstrating that HAPSTR1 loss is lethal even in mice lacking p53. This agrees with additional cell-line data we have added which indicate that p53 explains some, but not all, of the phenotypic effects of HAPSTR1 loss.

To your specific question about partial HAPSTR1 loss, heterozygous KO mice do not have similar phenotypes to the iKO mice. This is because the iKO model creates mosaic KOs (ie, tissues with full KO cells and WT cells), not heterozygous KO throughout the animal. We find that one copy of Hapstr1 distributed across all cells is sufficient for healthy life in mice (Figure added), but that widespread creation of knockout cells throughout tissues is poorly tolerated.

This is ultimately a difficult manuscript to review, as there are numerous interesting discoveries made. I would like to see some type of resolution for one of the three above questions prior to publication. Some additional depth to accompany the breadth of this study. There are many suggestions to explain these various results discussed in the manuscript, but some experiments to probe these models deeper would significantly improve this current manuscript. That being said, improving some of the experiments describing the regulation of HAPSTR1 by USP7 and TRIP12 (reps/quants/stats) could strengthen this paper sufficiently to publish in LSA as 'descriptive data'.

We appreciate the note that this paper makes multiple discoveries. We have added data and clarifications as discussed above.

Additional Points.

Figure 2d. No endogenous HAPSTR1 in nuclear fraction, is that because of detection? Also, nuclear localization of HUWE1 is not the most convincing in this figure.

We apologize for the image quality. We and others (see PMID: 37167062) have found that blotting the nuclear fraction for these endogenous proteins with the antibodies that are currently available and specific tends to be difficult. Our experience has been that, even with rapid fractionation protocols, a

large amount of nuclear HAPSTR1 (by immunofluorescence) tends to relocalize to cytosolic fractions. Additionally, we have attempted immunofluorescence for endogenous HAPSTR1 and HUWE1 to add to our immunoblot and mass spec data, but unfortunately none of the four antibodies tested showed specificity. Regarding nuclear HUWE1, this is a representative image for a consistent effect (more in nucleus with HAPSTR1 overexpression, less with knockdown) and best quality we can achieve with the reagents we have at hand.

Figure 4d needs quantification across different conditions to convince more efficiently.

We have updated the quantifications to reflect the mean of 3 independent time-course experiments.

Page 11 second paragraph. I think the Figure 3D callout should be Figure 4D.

Thank you we have fixed this typo.

February 15, 2024

RE: Life Science Alliance Manuscript #LSA-2023-02370-TR

Dr. Marc L Mendillo
Northwestern University
303 E. Superior St., Room 7-303
Chicago, Illinois 60610

Dear Dr. Mendillo,

Thank you for submitting your revised manuscript entitled "Tight regulation of a nuclear HAPSTR1-HUWE1 pathway essential for mammalian life". We would be happy to publish your paper in Life Science Alliance pending final revisions necessary to meet our formatting guidelines.

- please be sure that the authorship listing and order is correct
- Please upload your main and supplementary figures as single files. Do not include figure captions in these files.
- Please move the Figure legends (both main and supplementary) to the end of manuscript file (after the 'References' section)
- Please add a Summary Blurb/Alternate Abstract in our system
- Please add Keywords for your manuscript in our system
- Please add the Twitter handle of your host institute/organization as well as your own or/and one of the authors in our system
- Please add a conflict of interest statement to your main manuscript text
- Figure S5 panel D is missing from the figure. Please adjust the labels and callouts for this figure in the manuscript text accordingly.
- Figure S4 has only one panel, so the label of "a" should be removed from the figure and legend
- Please add a callout for Figures 5a, 6c, 6d, S3b, S3c, S4a to your main manuscript text; Callout for Figure 5i is present in the manuscript text while this label is not present in the figure itself.
- Please add scale bars to Figure 7d, S2c
- The contributions selected for author Clara Peek in the system do not on their own constitute authorship. Please either update this information in the system and in the Author Contributions section of the manuscript, or let us know if the author should be removed.

A. FINAL FILES:

B. MANUSCRIPT ORGANIZATION AND FORMATTING:

Sincerely,

Reviewer #1 (Comments to the Authors (Required)):

The authors have addressed most of our questions.

Reviewer #2 (Comments to the Authors (Required)):

The authors have sufficiently addressed my concerns from the initial submission. While this manuscript still described a broad spectrum of results, I think that the community will be interested in the work and it will provide a nice springboard for this lab and others to continue pursuing HAPSTR1 biology.

February 27, 2024

RE: Life Science Alliance Manuscript #LSA-2023-02370-TRR

Dr. Marc L Mendillo
Northwestern University
303 E. Superior St., Room 7-303
Chicago, Illinois 60610

Dear Dr. Mendillo,

Thank you for submitting your Research Article entitled "Tight regulation of a nuclear HAPSTR1-HUWE1 pathway essential for mammalian life". It is a pleasure to let you know that your manuscript is now accepted for publication in Life Science Alliance. Congratulations on this interesting work.

DISTRIBUTION OF MATERIALS:

Again, congratulations on a very nice paper. I hope you found the review process to be constructive and are pleased with how the manuscript was handled editorially. We look forward to future exciting submissions from your lab.

Sincerely,
